# Glucose-6-phosphate-dehydrogenase on old peroxisomes maintains self-renewal of epithelial stem cells after asymmetric cell division

Hien Bui[1,2], Simon Andersson[1,2,8], Agustin Sola-Carvajal [3,8],
Tommaso De Marchi [4], Eliisa Vähäkangas[1,2,5], Minna Holopainen [1,6],
Andrew H. House[1,6], Bohdana M. Rovenko[1,2], Johanna I. Englund [2],
Maria Kasper [3], Emilia Kuuluvainen[1,2], Reijo Käkelä [1,6], Ville Hietakangas [1,2],
Emma Niméus [4,7] & Pekka Katajisto [1,2,3] ✉

Selective inheritance of sub-cellular components has emerged as a mechanism guiding stem cell fate after asymmetric cell divisions. Peroxisomes play a crucial role in multiple metabolic processes such as fatty acid metabolism and reactive oxygen species detoxification, but the apportioning of peroxisomes during stem cell division remains understudied. Here, we develop a mouse model and labeling technique to follow the dynamics of distinct peroxisome age-classes, and find that old peroxisomes are inherited by the daughter cell retaining full stem cell potency in mammary and epidermal stem cell divisions. Old peroxisomes carry Glucose-6-phosphate-dehydrogenase, whose specific location on the peroxisomal membrane promotes stem cell function by facilitating peroxisomal ether lipid synthesis. Our study demonstrates age-selective apportioning of peroxisomes in vivo, and unveils how functional heterogeneity of peroxisomes is utilized by asymmetrically dividing cells to metabolically divert the fate of the two daughter cells.

In addition to macromolecules like RNAs and transcription factors (reviewed in[1]), complete cellular components and organelles (e.g. mitochondria[2], lysosomes[3,4], and centrosomes[5,6]) can be asymmetrically segregated during cell division to influence cell fate. We previously found that in asymmetric cell division (ACD) of stem-like human mammary epithelial cells (hMECs), young mitochondria are apportioned to the daughter cell retaining self-renewal capacity[2].

Interestingly, the functional differences between mitochondrial age-classes alter the metabolism and redox state of their recipient cells[7], raising the question regarding other redox active organelles, such as peroxisomes. A previous study also suggested that peroxisomes regulate mitotic spindle orientation during ACD of epidermal stem cell (EpSC) and thereby also influence cell fate decisions[8]. However, whether peroxisomes themselves are selectively segregated during ACD,

[1]Molecular and Integrative Biosciences Research Programme, Faculty of Biological and Environmental Sciences, University of Helsinki, 00790 Helsinki, Finland. [2]Institute of Biotechnology, Helsinki Institute of Life Science (HiLIFE), University of Helsinki, 00790 Helsinki, Finland. [3]Department of Cell and Molecular Biology, Karolinska Institutet, 17177 Stockholm, Sweden. [4]Division of Oncology and Surgery, Department of Clinical Sciences, Lund University, 22362 Lund, Sweden. [5]Stem cells and metabolism research program, Faculty of Medicine, University of Helsinki, 00014 Helsinki, Finland. [6]Helsinki University Lipidomics Unit (HiLIPID), Helsinki Institute of Life Science (HiLIFE), and Biocenter Finland, University of Helsinki, 00014 Helsinki, Finland. [7]Department of Surgery, Skåne University Hospital, 22242 Lund, Sweden. [8]These authors contributed equally: Simon Andersson, Agustin Sola-Carvajal. ✉e-mail: pekka.katajisto@helsinki.fi

and whether peroxisomal functions can influence cell fate remains an open question.

## Results

### Bipotent basal MECs receive old peroxisomes in ACD

To start analyzing age-selective apportioning of mature peroxisomes in ACD, we generated hMECs expressing a peroxisome targeted SNAPtag[9] (SNAP-PTS1), and labeled peroxisomes sequentially to distinguish old and young peroxisomes (enriched for SNAPtags generated either 23-33 hours, or ≤6 hours before the analysis, respectively) with cell permeable fluorescent SNAPtag substrates[2] (Fig. 1a). Strikingly, a subset of cells apportioned peroxisomes age-selectively (Fig. 1a). However, opposite to our earlier findings on mitochondria[2],

hMECs inheriting older peroxisomes formed more mammospheres, indicating higher self-renewal capacity (Supplementary Fig. 1a). To address the potential role of age-selective apportioning of peroxisomes in primary cells, we developed a mouse model where a Lox-Stop-Lox-SNAPtag-PTS1 construct is introduced to the R26 locus (SNAP-PTS1 mouse) (Fig. 1b). Mice expressing the peroxisomal SNAP-tag ubiquitously (PGK-Cre;SNAP-PTS1) developed normally with a strictly peroxisome specific localization of the SNAPtags (Supplementary Fig. 2a–c). We isolated basal and luminal primary Mammary Epithelial cells (mMECs) by fluorescence activated cell sorting (FACS), and cultured them separately in vitro followed by sequential labeling of old and young peroxisomes that was adjusted to mark peroxisomes with SNAPtags imported >51 hours and <5 hours ago, respectively

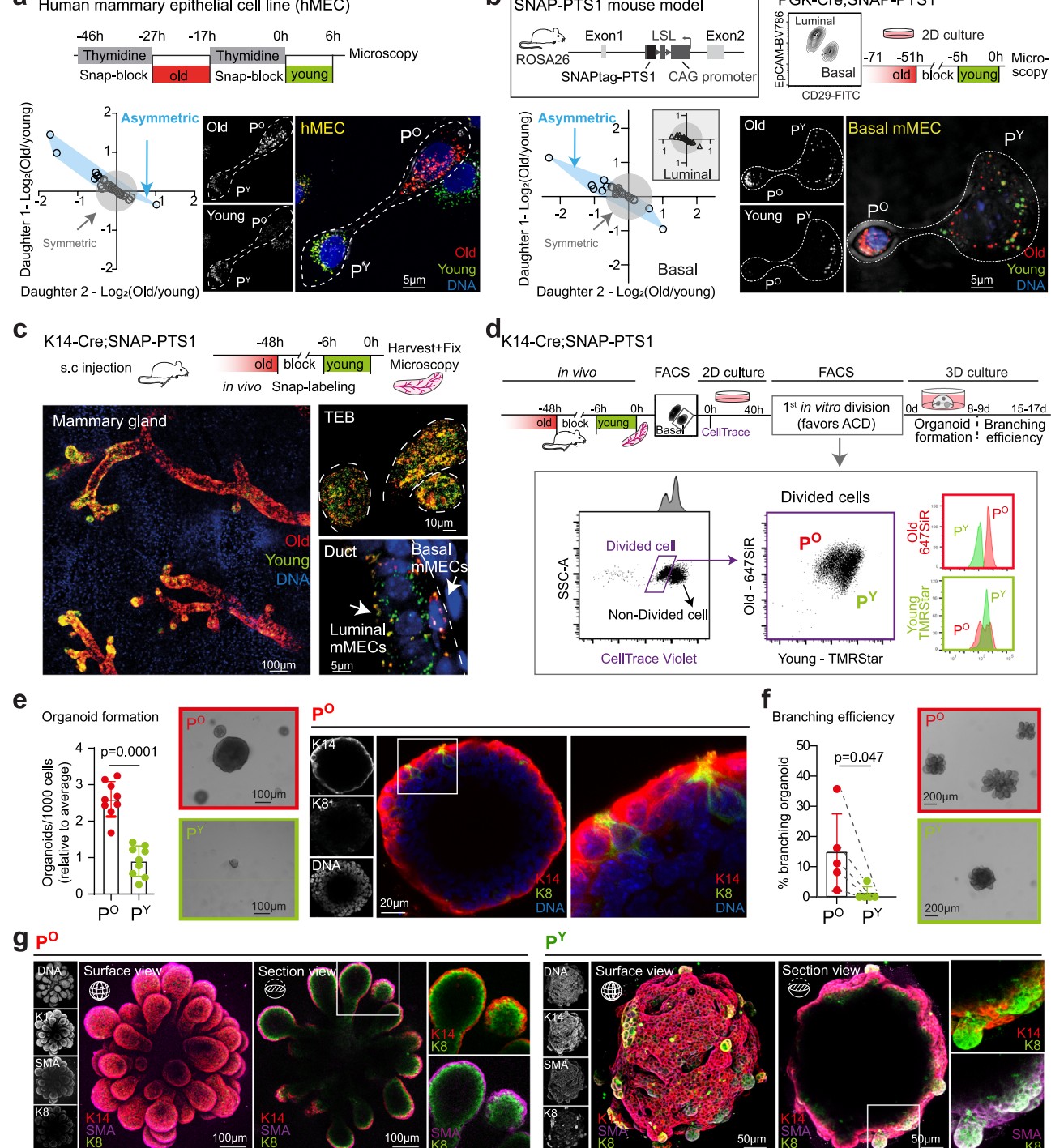

**Fig. 1 | (caption continues on following page)**

**Fig. 1 | Old peroxisomes are inherited by the bipotent stem cell in asymmetric division of mouse mammary epithelial basal cells. a** SNAP labeling and imaging of old and young peroxisomes in synchronously dividing hMECs. Representative image and quantification of age-selective (symmetric and asymmetric) apportioning of old and young peroxisomes between daughter cells in cell division. Each data point represents a cell division with x- and y-axis showing the ratio of old/young peroxisomes in daughter cells. Data from 86 cell division pairs, collected from four independent experiments. **b** Schematic of knock-in strategy for inducible SNAP-PTS1 mouse (black box), and FACS isolation of basal and luminal mMECs and in vitro labeling of peroxisomal age classes for analysis of age-selective apportioning of peroxisomes in cell division. Age-selective apportioning of peroxisomes occurs in basal cells (blue) but not luminal cells (gray, right conner). Each datapoint represents a daughter cell pair during cytokinesis. Representative image of a basal mMEC is shown. Data from 31 pairs of basal cells and 24 pairs of luminal cells, collected from three independent experiments. **c** In vivo labeling of different age classes of peroxisomes in mammary glands of K14-Cre;SNAP-PTS1 mice by injecting SNAP substrates into mammary glands. TEB: Terminal end-bud. **d** Schematic of in vivo and ex vivo labeling of peroxisomal age classes in mammary glands, and of the functional assays on daughter cells inheriting old ($P^O$) or young ($P^Y$) peroxisomes of the mother cell in basal mMEC asymmetric cell divisions (ACD). Cells that had divided were analyze for enrichment of old and young peroxisomes, revealing the $P^O$ and $P^Y$ populations (gray box). **e** Organoid forming capacity of $P^O$ and $P^Y$ daughter cells emerging from the first basal mMEC cell division in vitro. Immuno-fluorescent staining of an organoid originating from a $P^O$ cell shows K14 positive basal cells and K8 positive luminal cells. Data from nine biological replicates, $p$-value from two tailed t-test. **f** Analysis of branching efficiency of the organoids from (**e**). Organoids originating from $P^O$ cells branch with a significantly higher frequency than the organoids originating from $P^Y$ cells. Data from five biological replicates, $p$-value from two tailed t-test. **g** Representative immunofluorescence images of a $P^O$ branching organoid with a bi-layer structure resembling organization of the mammary epithelium in vivo, and a $P^Y$ organoid with disorganized structure. Data are presented as mean ± SD. Source data are provided as a Source Data file.

(Fig. 1b). While total peroxisomal content of the mother cell was asymmetrically apportioned in subset of both luminal and basal cell divisions (Supplementary Fig. 2d), age-selective apportioning was noticed intriguingly only in the basal mMECs (Fig. 1b). This observation interestingly aligned with the previous reports that only basal cells give rise to the two mammary epithelial lineages[10], prompting questions about the role of age-selective peroxisome distribution in determination of cell fate and lineage.

Therefore, we next focused on the age-selective peroxisome apportioning in basal cells. To probe its functional consequences, we switched to K14-Cre;SNAP-PTS1 mice, where SNAPtag expression is induced during embryogenesis and occurs in both luminal and basal cells of the adult mammary gland (Fig. 1c). Sequential injection of SNAP substrates into adult mammary glands revealed heterogeneity in peroxisomal age within the epithelial cells, especially in the terminal end-buds (Fig. 1c). Interestingly, basal cells were in general enriched with old peroxisomes and luminal cells had younger peroxisomes (Fig. 1c), despite similar proliferation frequency (Supplementary Fig. 2e). We next exploited the ability of isolated basal progenitors to regain bipotency and give rise to both basal and luminal lineages upon in vitro culture[10–14]. We labeled old peroxisomes in vivo and young peroxisomes ex vivo during basal cell isolation, and plated cells for 2D culture (Fig. 1d). Importantly, the first in vitro division by primary basal mMECs is known to favor ACD[15], and we identified cells that had completed their first division by dilution of the CellTrace-dye (Fig. 1d). We then re-isolated daughter cells that had received either high or low portion of the mother cells old peroxisomes ($P^O$ and $P^Y$ respectively in Fig. 1d) and assessed their self-renewal potential by seeding them in Matrigel for 3D organoid culture[16]. Strikingly, $P^O$ cells formed three times more organoids than $P^Y$ cells, indicating higher self-renewal capacity (Fig. 1e). Moreover, when staining these organoids with keratin 14 (K14) and keratin 8 (K8), the markers for basal and luminal cells in vivo respectively, they contained few cells positive for the luminal cell marker keratin 8 (K8) (Fig. 1e). As the organoids originated from K14 positive basal cells (Fig. 1d), and we detected no contaminating luminal cells in the original sorted basal cell population (0/180 cells K8 +, Supplementary Fig. 2f), these data suggested that some daughters of basal cells may indeed regain bipotency in our assay.

To more formally assess bipotency of $P^O$ and $P^Y$ daughters, we administered FGF2 to organoids to induce branching morphogenesis. As branching requires balanced generation of both basal and luminal cells[10,17], branching of organoids initiated from a single mammary epithelial cell lineage indicates bipotent progenitor activity (Supplementary Fig. 2g). Organoids from $P^O$ daughter cells branched with a significantly higher frequency than the organoids that originated from $P^Y$ daughter cells (Fig. 1f). Moreover, the branched $P^O$ organoids formed a bi-layer structure where cells positive for K14 and smooth muscle actin (SMA) formed the outer basal layer, and K8 positive cells

were contained as the inner luminal layer, resembling the in vivo mammary gland organization[11,12] (Fig. 1g). In contrast, the non-branching $P^Y$ organoids had a disorganized morphology with over-lapping expression of luminal and basal markers, and K8 positive cells at the outer layer. Interestingly, such $P^Y$ organoids were quite similar to the organoids formed by primary luminal cells (Fig. 1g, Supplementary Fig. 2h), suggesting the $P^Y$ cells had acquired luminal traits in ACD - including loss of bipotency. Importantly, the difference in organoid forming capacity and branching efficiency between $P^O$ and $P^Y$ daughter cells was not dependent on the total quantity of old peroxisomes in the mother cell prior to the division (Supplementary Fig. 2i) indicating that the differences emerge during ACD in vitro. Taken together, these results indicated that in ACD, the basal mMEC daughter that receives older peroxisomes is not only more self-renewing, but can also produce two cell lineages in a balanced fashion that indicates bipotency.

## G6PD on old peroxisomes maintains stem cell fate by promoting peroxisomal metabolism

We next set out to investigate how selective inheritance of perox-isomal age-classes influences the cellular fate and potency of daughter cells. We adapted FACS to single organelle sorting to separate old and young peroxisomes for mass spectrometry (MS). Due to limited amount of mMECs, we used hMECs for this approach. Briefly, we labeled old and young peroxisomes in intact cells and isolated per-oxisomes using differential density gradient centrifugation with iodixanol layering (Fig. 2a, Supplementary Fig. 3a), and used a mito-chondrial dye (MitoTracker) to exclude mitochondria. However, events positive for both peroxisomal SNAP-label and MitoTracker were considered to represent potential interactions between the two orga-nelles (Pero$^{MitoTr+}$) (Supplementary Fig. 3b, c).

We found 68 consistently detected peroxisome associated pro-teins, which constitutes 35% of peroxisome-associated proteins (68/196)[18] (Supplementary Fig. 3d, Supplementary data 1). While the low representation of peroxisomal proteins in our age-selectively sorted samples (Supplemental data 1) raises the concern about con-taminating proteins, the Snap-label based sorting enriched perox-isomal proteins in comparison to the pre-sorting peroxisomal sample (Supplementary Fig. 4a, Supplementary data 2). In addition, the 68 detected peroxisomal proteins include both membrane and matrix associated proteins indicating peroxisomes remain intact during sorting. Beside peroxisomal proteins, we also detected proteins of other cellular compartments in all the samples (Supplementary Fig. 4a, Supplementary data 1,2). Intriguingly, the Pero$^{MitoTr+}$ were particularly enriched with proteins from other organelles in addition to mito-chondrial proteins (Supplementary Fig. 4a,b), suggesting that they may represent cellular compartments active in inter-organelle com-munication. However, as our experimental setup limits quantity of the analyzed samples, it is likely that not all found proteins represent bona

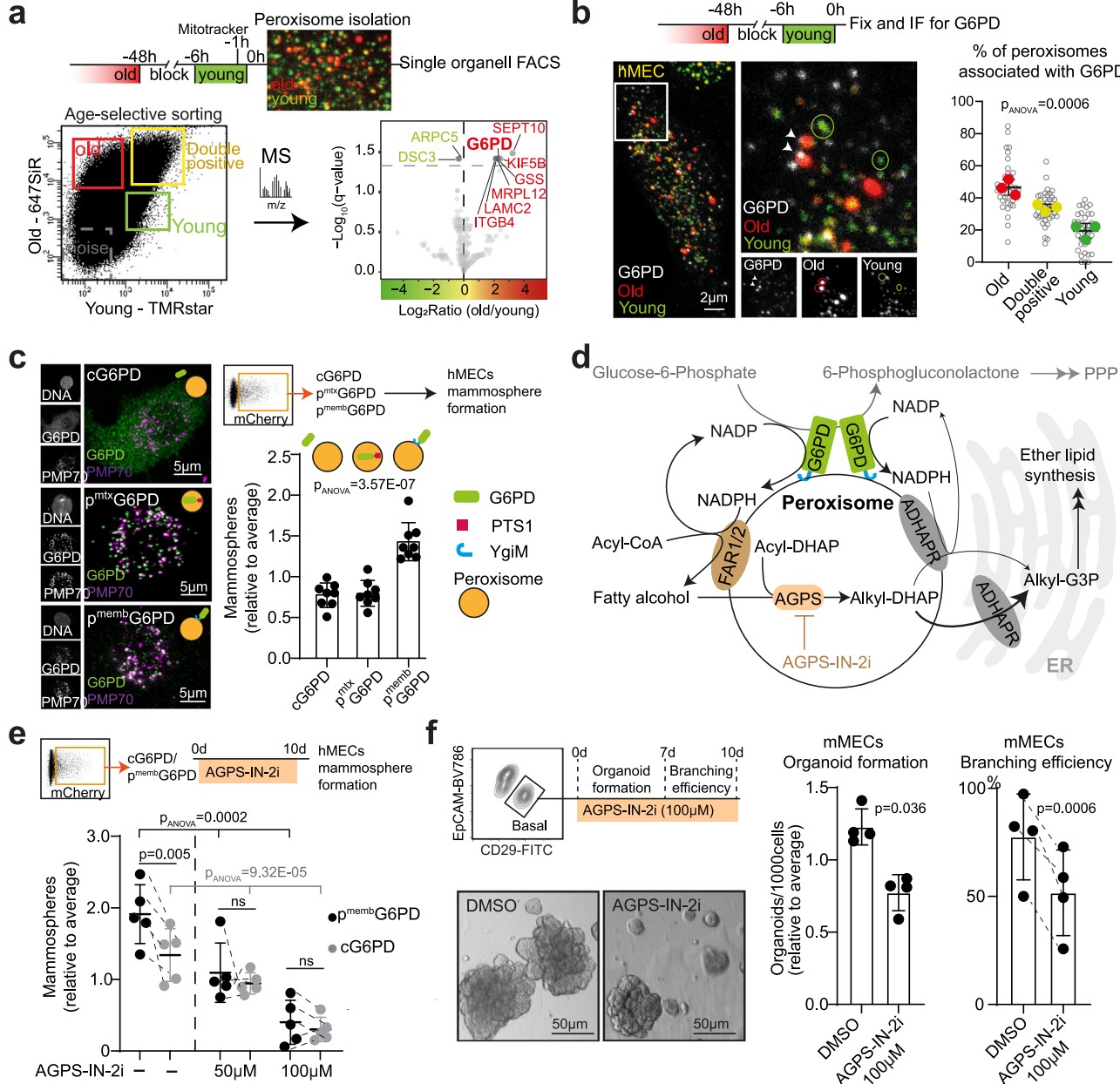

**Fig. 2 | G6PD is enriched in old peroxisomes and influences the fate of daughter cells after asymmetric cell division. a** Age-selective isolation of peroxisomes and their proteomic analysis using mass spectrometry. Volcano plot of proteins differentially enriched between old and young peroxisomes. Significantly enriched proteins are marked in red (enriched with old peroxisomes) and green (enriched with young peroxisomes). **b** Immunofluorescent labeling of G6PD in hMECs with age-selectively labeled peroxisomes shows colocalization of G6PD with peroxisomes. G6PD is significantly enriched on old peroxisomes. Data shows 38 individual cells (unfilled circles) from three independent experiments (filled circles), *p*-value from one-way ANOVA test of independent experiments. **c** Representative immunofluorescence images show localization of G6PD (green) over-expressed in cytosol (cG6PD), in the peroxisomal matrix (p^mtx^G6PD) using a PTS1 targeting signal, and on the peroxisomal membrane (p^memb^G6PD) using a YgiM tail anchor. PMP70 immunofluorescent labeling show peroxisomes. Overexpression of G6PD on the peroxisomal membrane, but not in the peroxisomal matrix or in the cytosol,

increases mammosphere formation of hMECs significantly. Data from eight independent experiments, *p*-value from one-way ANOVA test. **d** Schematic showing the G6PD activity and peroxisomal ether lipid synthesis pathway. NADPH produced by G6PD activity on the peroxisomal membrane is potentially utilized for peroxisomal ether lipid synthesis, and inhibiting AGPS can be employed as a strategy to probe the function of p^memb^G6PD on peroxisomal ether lipid synthesis. **e** Mammosphere formation by cG6PD and p^memb^G6PD after continuous inhibition of AGPS. AGPS inhibition reduces mammosphere formation and mitigates the boost induced by p^memb^G6PD. Data from five experiments, *p*-value from paired, two tailed *t*-test and one-way ANOVA test. **f** Organoid formation and branching efficiency of mMECs after continuous inhibition AGPS activity. Inhibiting peroxisomal ether lipid synthesis reduces organoid formation and branching efficiency of mMECs. Data from four biological replicates, *p*-value from paired, two tailed *t*-test. Data are presented as mean ± SD. Source data are provided as a Source Data file.

fide peroxisomal proteins. Consequently, the data set produced by this experiment, which combined traditional peroxisomal isolation with single-organelle sorting, should not be considered as a reference for peroxisomal proteome - but rather a comparison between recently generated and older peroxisomes.

In general, old and young peroxisomes were relatively similar, differing significantly only in the levels of 9 proteins (Fig. 2a). Among these, Glucose-6-phosphate-dehydrogenase (G6PD) presented the singular exception that has direct links with the peroxisomal functions, such as lipid, ROS, and NADP metabolism (Supplementary Fig. 4c).

Although G6PD is primarily localized in cytoplasm, in line with some earlier works[18–20], we found that a subset of the abundant G6PD indeed associates with peroxisomes (Fig. 2b). Furthermore, G6PD preferentially associated with old peroxisomes, confirming our proteomic analysis (Fig. 2b). To test whether the enrichment of G6PD on old peroxisomes is necessary for self-renewal capacity after ACD of mMECs, we inhibited G6PD activity with a small molecule (G6PDi-1)[21] transiently after ACD, and analyzed the self-renewing capacity of daughter cells. Excitingly, G6PDi-1 reduced organoid forming capacity of the $P^O$ daughter cells, but had no impact on $P^Y$ daughter cells (Supplementary Fig. 5a, b).

G6PDi-1 is expected to inhibit G6PD throughout the cell, and we noted that age-selective asymmetric segregation of peroxisomes did not result in asymmetric apportioning of total G6PD (Supplementary Fig. 5c). We therefore probed the specific role of peroxisomal G6PD by over-expressing G6PD in hMECs in three locations: in the cytosol (cG6PD), in the peroxisomal matrix ($p^{mtx}$G6PD) using PTS1, and on cytoplasmic face of the peroxisomal membrane ($p^{memb}$G6PD) with the YgiM tail anchor targeting[22] (Supplementary Fig. 5d). Overexpressing cells were analyzed for mammosphere forming capacity (Fig. 2c). Excitingly, $p^{memb}$G6PD increased the mammosphere formation, whereas targeting to the other two locations had no impact (Fig. 2c, Supplementary Fig. 5e, f).

G6PD is an important enzyme in both the pentose phosphate pathway (PPP)[23] and cellular redox balance activities[24]. As we previously found that high PPP activity is necessary for maintaining stemness in the daughter cells inheriting young mitochondria during ACD[7], we checked weather $P^O$ had higher PPP activity than $P^Y$. By analyzing the ratio of M + 1 and M + 2 labeled lactate[25] in $P^O$ and $P^Y$ cells that were cultured in media containing 1,2$^{13}$C glucose for one hour after ACD, we found $P^O$ to indeed have high PPP/glycolysis -ratio (Supplementary fig. 5g). However, none of the three G6PD over-expression strategies increased the lactate M + 1/M + 2 ratio significantly (Supplementary Fig. 5h), suggesting that other enzymes are responsible for the self-renewal promoting PPP activity[7,26]. Additionally, supplementing hMECs with the antioxidant glutathione, GSH-ME, did not mitigate the impact of G6PDi-1 on mammosphere formation (Supplementary Fig. 5i), indicating that G6PD does not primarily maintain stemness via cellular redox balance. Collectively, these data suggested that the localization of G6PD dehydrogenase activity on the peroxisomes may have local effects on the peroxisomal functions.

G6PD consumes Glucose-6-phosphate and NADP to generate 6-phosphogluconolactone and NADPH. NADPH is involved in multiple peroxisomal metabolic pathways, including, fatty acid oxidation, ROS management by Glutathione, and ether lipid biosynthesis (reviewed in[27]). Interestingly, peroxisomal ether lipid synthesis depends primarily on two NAPDH-dependent enzymes residing on the peroxisomal membrane, namely Fatty acyl-CoA reductases 1 and 2 (FAR1 and FAR2)[27–30] (Fig. 2d). To address whether peroxisomal G6PD promotes ether lipid synthesis, we analyzed the lipidome of cG6PD and $p^{memb}$G6PD expressing hMECs. While the total lipid content, or the lipid and fatty acid class compositions were not influenced (Supplementary fig. 6a), stearic acid (FA 18:0) was significantly reduced in $p^{memb}$G6PD cells, accompanied by around 10% increase in the downstream ether lipid PE P-36:1 (Supplementary fig. 6b). To address if the increased peroxisomal ether lipid synthesis contributes to stemness maintenance, we used AGPS-IN-2i[31] to inhibit Alkylglycerone-Phosphate Synthase (AGPS). AGPS acts in the peroxisomal matrix, converting fatty alcohol - produced by FAR1/2 activity on the peroxisomal membrane - into alkyl-DHAP (Fig. 2d). Importantly, inhibition of AGPS at a dose that had no impact on normal 2D growth of cells (Supplementary Fig. 6c) reduced the mammosphere forming capacity of hMECs and mitigated the boost induced by $p^{memb}$G6PD (Fig. 2e). AGPS-IN-2i also reduced the formation and branching of organoids from primary mMECs (Fig. 2f, Supplementary Fig. 6d, e). Taken together, these data

demonstrate that G6PD on the old peroxisomes contributes to fate determination after ACD via local effects at the peroxisomal membrane, and bring forward the important role of peroxisomal ether lipid synthesis in stem cell maintenance.

## Age-selective segregation of peroxisomes in ACD in vivo

Finally, we asked whether age-selective apportioning of peroxisomes also occurs in vivo. As unperturbed mammary epithelial cells do not divide asymmetrically after birth, we focused on epidermal stem cells (EpSCs) that can divide either symmetrically or asymmetrically in the early postnatal skin[32]. K14-Cre drives SNAPtag-PTS1 expression also in the epidermal basal cells (Fig. 3a), and the mitotic spindle angle parallel (0°-10°) with the basement membrane (BM) is considered to mark symmetric divisions, whereas perpendicular angle (60°–90°) is associated with asymmetric cell division in the interfollicular EpSCs[32]. By examining epidermis after in vivo subcutaneous SNAP substrate administration, we found that the perpendicularly dividing cells apportioned older peroxisomes to the daughter cell that remained at the BM and is therefore expected to maintain stemness (Fig. 3b). While majority of the parallel divisions apportioned peroxisomes symmetrically, a small subpopulation interestingly apportioned peroxisomes as age-selectively as perpendicular divisions (Fig. 3b). As previous reports indicate that a subset of such parallel-to-BM EpSC divisions can also give rise to daughter cells with asymmetrically predetermined fates[33–35], we analyzed the organoid forming capacity[36] of CD49f$^{high}$;Sca1$^+$ basal cells that were enriched either for old ($P^O$) or young ($P^Y$) peroxisomes in vivo (Fig. 3c, Supplementary fig. 7a). In line with our findings on in vitro dividing mammary cells, $P^O$ basal EpSCs formed more epidermal organoids than $P^Y$ cells (Fig. 3c). Moreover, while gross morphology of $P^Y$ and $P^O$ epidermal organoids was similar, $P^O$ derived organoids had generated a more robust pool of suprabasal cells positive for Keratin10 (K10) in addition to maintaining a well-defined pool of K14+ basal cells (Fig. 3d, Supplementary fig. 7b). Altogether, these data indicate that old peroxisomes are inherited by the stem cell daughter during asymmetric divisions of EpSCs in vivo.

Further recapitulating the findings in mammary cells, G6PD inhibition reduced the organoid forming and differentiation potency of $P^O$ EpSCs but had no effect on $P^Y$ cells (Fig. 3e, Supplementary fig. 7c). And finally, inhibiting AGPS activity reduced organoid formation of EpSCs (Fig. 3f, Supplementary Fig. 7d). Taken together, these data show that age-selective apportioning of peroxisomes occurs in epithelial stem cell divisions, and that peroxisomal ether lipid synthesis is important for maintaining stemness across epithelial tissues.

## Discussion

Asymmetric cell divisions have been mostly studied in non-mammalian systems due to technical challenges, which may bias estimates on ACD frequency in mammalian tissues. The mouse model developed in this study allows age-specific labeling and follow-up of organelle dynamics in ACD. Using in vivo organelle labeling, our study provides evidence demonstrating that mammalian stem cells apportion old peroxisomes qualitatively and age-selectively to the daughter cell destined to become the new stem cell in ACD in vitro and in vivo. Our results also highlight the ability of age-selectively inherited organelles to influence cell fate regardless of the extrinsic factors, but this does not exclude a potential involvement of intrinsic mechanisms such as the Par3-Par6-aPKC polarity complex and the NumA-LGN-Gαi cortical complex (reviewed in[35,37–40]).

G6PD is suggested to influence stem cell functions by regulating PPP and cellular Redox balance (reviewed in[41,42]). While we cannot exclude the existence of additional mechanisms, we here discover the role of typically cytoplasmic G6PD specifically on peroxisomal metabolism and how it regulates cell fate through ether lipid biosynthesis. This reveals spatially compartmentalized metabolism as a new level of stem cell regulation, and highlights potential of peroxisomal functions

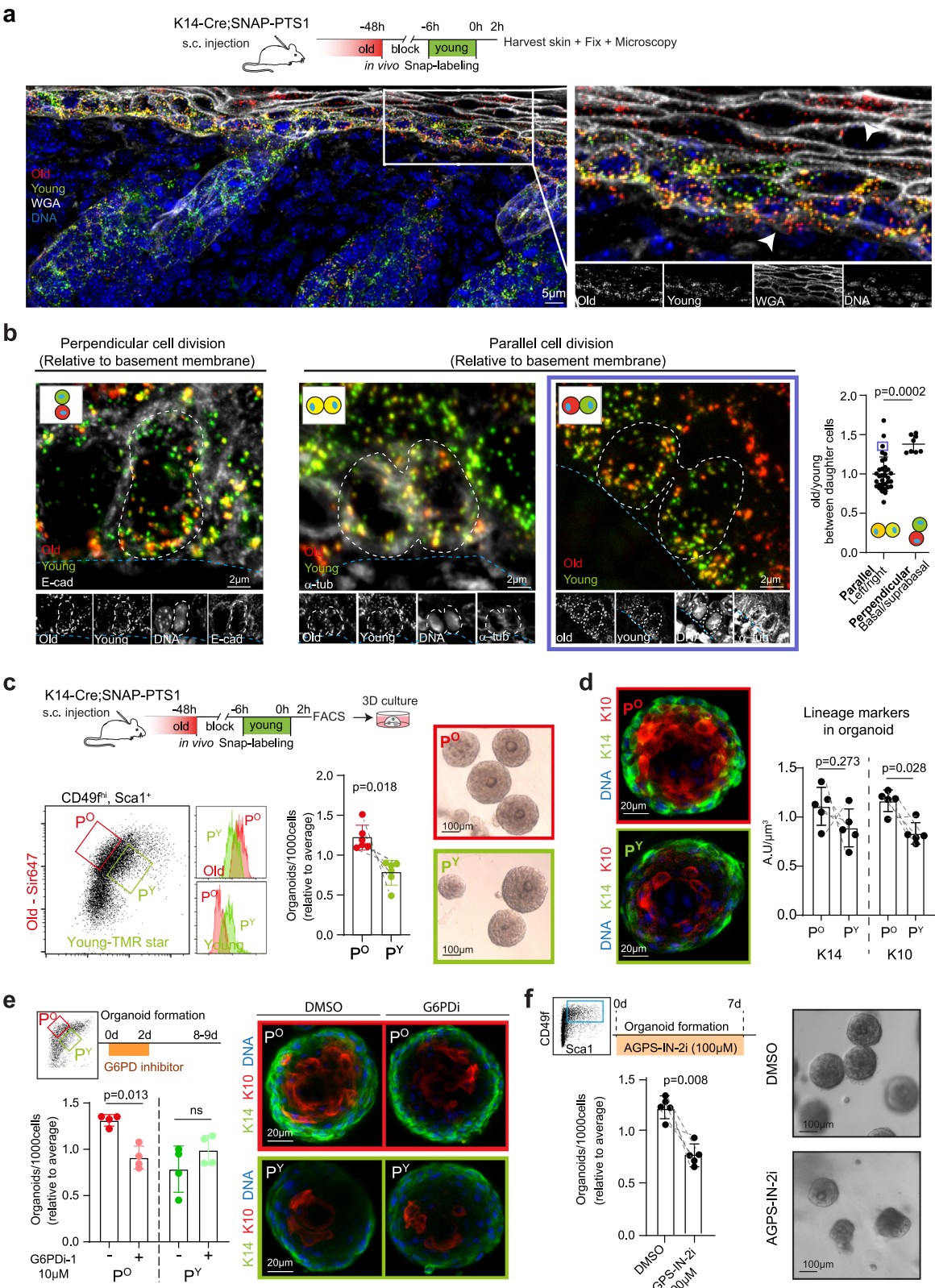

that are particularly dependent on links with other subcellular compartments (reviewed in[43]). Ether lipids influence membrane organization and dynamics, cellular signaling, and oxidative stress, and they are implicated in neurodegeneration, aging, and cancer (reviewed in[44]). Whether stem cell regulation by peroxisomal ether lipid synthesis contributes to such conditions is currently not known. Consequently, further studies on how G6PD is recruited to peroxisomes, especially older ones, and how peroxisomal ether lipid synthesis affects stem cell functions are crucial for advancing our understanding of these processes.

Peroxisomes are abundant and typically dispersed in mammalian cells, leading to the assumption that they are randomly distributed during cell division. However, some studies have shown that peroxisomes can aggregate at mitotic spindle poles, either to ensure their

**Fig. 3 | Age-selective segregation of peroxisomes in asymmetrically dividing basal skin cells in vivo. a** Schematic of in vivo labeling of different age classes of peroxisomes in skin of K14-Cre;SNAP-PTS1 mice by injecting SNAP substrates subcutaneously. Representative images show the heterogeneity of peroxisomal ages in tissue. Note the enrichment of old label in the basal cells, and in the cells forming the outer layers of skin (white arrow heads). Wheat germ agglutinin (WGA) labels the cell membrane. **b** Representative images and quantification of age-specifically labeled peroxisomes in basal skin cells at cytokinesis in vivo. Cells dividing perpendicular to the basement membrane allocate old peroxisomes to the daughter cell staying at the basement membrane (left). Majority of cells dividing parallel to the basement membrane apportion peroxisomes symmetrically (middle image), but some apportion peroxisomes age-selectively (right, with blue border). Graph shows old/young peroxisomes ratio between the two daughter cells, classified by position of the two daughter cells relative to the basement membrane. Data shows 42 cell division pairs, from five biological replicates, *p*-value from two tailed t-test of biological replicates. **c** Schematic of in vivo labeling of peroxisomal age classes in skin, FACS sorting strategy and the functional assays on cells enriched with old (P$^O$) or young (P$^Y$) peroxisomes in CD49f$^{hi}$, Sca1$^+$ EpSCs. Left panels show representative FACS data. Middle panels demonstrate that P$^O$ has a higher organoid-forming capacity compared to P$^Y$. Right images show phase contrast images of organoids from P$^O$ or P$^Y$. Data from six biological replicates, *p*-value from paired, two tailed t-test. **d** Immunofluorescent staining of organoids originating from a P$^O$ or P$^Y$ cells and quantification of lineage markers in these organoids. Organoids from P$^O$ have significantly higher levels of the suprabasal differentiation marker, K10, than organoids from P$^Y$. Data from five biological replicates, *p*-value from paired, two tailed t-test. **e** Organoid formation by P$^O$ and P$^Y$ EpSCs after transient inhibition of G6PD. G6PD inhibition reduces organoid forming capacity of P$^O$. Data from four biological replicates, *p*-value from paired, two tailed t-test. **f** Organoid formation of EpSCs after continuous inhibition AGPS activity. Inhibiting peroxisomal ether lipid synthesis reduces organoid formation of EpSCs. Data from five biological replicates, *p*-value from paired, two tailed t-test. Data are presented as mean ± SD. Source data are provided as a Source Data file.

inheritance by daughter cells[45] or to influence cell fate via spindle orientation[8]. These findings, along with our results on the functions of old peroxisomes in epithelial stem cell maintenance, suggest that peroxisomal segregation during cell division may be more tightly regulated than previously assumed. Interestingly, the proteins identified as key factors in this process are primarily peroxisomal biogenesis factors, such as Pex11b, Pex14[8], and Pex19[46], rather than those associated with cytoskeleton or spindle poles. This indicates that further studies are needed to fully understand the involved mechanisms, and highlights the potential of our age-selective labeling system for investigating this process.

It is also important to note that the labeling strategy used here to separate subsets of peroxisomes according to their chronological "age", is likely not capturing the complete heterogeneity of peroxisomes. Moreover, peroxisomal turnover and biogenesis are dynamically regulated, and we used only one timing scheme to separate peroxisomal age-classes. Even with these limitations, we discover great heterogeneity in organellar age among cells within tissues (Figs. 1c, 3a), suggesting that, in addition to asymmetric division, organelle age may impact tissue dynamics and cellular functions more broadly.

## Methods
### Mice strains and housing
Animal housing and experiments were done in accordance to Finnish National Animal Experimentation Board with ethical approval. All mice were maintained in a C57BL/6 background and housed in IVC cages with 12 hours light-dark cycle, at 22 °C, with 50-70% humidity and ad libitum access to food (2916, Inotiv).

The SNAP-PTS1 mouse (MGI:6466977) was generated by introducing a CAG-LoxP-3*polyA-loxP- SNAPtag-PTS1 cassette into the first intron of ROSA26 in reverse direction by homologous recombination. To induce the expression of SNAPtag-PTS1 in all the tissues, SNAP-PTS1 mice were crossed with PGK-Cre mice[47]. For tissue specific expression in epidermal stem cells (EpSCs) and mammary epithelial cells (mMECs), SNAP-PTS1 female were crossed with K14-Cre male mice[48].

### Cell lines and culture
Human mammary epithelial cells (hMECs) line FL2[49] expressing SNAPtag-PTS1 (hMEC SNAP-PTS1) were generated by lentiviral infection. The SNAPtag-PTS1 construct (Addgene, #182883) was generated by replacing the OMP25 sequence of the of Snap-Omp25 plasmid[2] (Addgene #69599), with the peroxisomal targeting signal 1 (PTS1)[50]. Briefly, the Snap-Omp25 plasmid was restricted with EcoRI and MluI enzymes (New England Biolabs, R3101S, R3198S) to remove the OMP25 sequence. The digested plasmid was purified using GeneJET Gel Extraction Kit (Thermo Fisher Scientific, K0691). Two single complementary stranded oligonucleotides containing PTS1, Mlu and EcoRI

restriction sequences were annealed at 95 °C for 5 minutes, 25 °C for 1 hour in T4 ligation buffer (NEB, B0202S), followed by ligation with the digested plasmid using T4 DNA ligase (NEB, M0202S) according to the manufacturer's protocol. Lentiviruses were produced in HEK293FT cells (Invitrogen, R70007) by transfecting cells with targeting plasmid (SNAPtag-PTS1), packaging plasmid pCMV-dR8.91 and envelop plasmid VSV/G using jetPEI (Polyplus transfection, 101-10 N) according to the manufacturer's instructions. Media containing viruses was collected after 2-3 days and used for cell infection.

Sequence of the oligonucleotides (5'-3'):
PTS1-Mlu-F: CGCGTAAGAGCAAGCTGTGAG
PTS1-EcoRI-STOP-R: AATTCTCACAGCTTGCTCTTA

hMECs were maintained in MEGM mammary epithelial medium (Lonza, CC-3153) as described before[2]. For live imaging, MEBM was replaced by phenol-red free MEBM (Lonza, CC-3150).

### In vivo SNAP labeling
SNAP substrates (all from New England Biolabs, see below) were dissolved in 6 μl DMSO and diluted with saline to final concentration at desired volume.

For in vivo mammary glands labeling: 240 μl of labeling solution (15 nmols SNAP-Cell 647-Sir (S9102S), 25 nmols SNAP-Cell Block (S9106S), 15 nmols SNAP-Cell TMR-Star (S9105S)) was injected subcutaneously into mammary glands of a mouse at 4 positions within the area formed by mammary gland pair 2-3 (left/right pectoral) and 4-5 (left/right inguinal). Timeline of injections is presented in figures. For organoid forming and branching experiments (Figs. 1 and supplementary fig. 5), SNAP-Cell Sir647, SNAP-Cell Block were injected to mammary glands as above except for SNAP-Cell TMR-star, which was added to collagenase solution during MECs isolation.

For in vivo skin labeling: post-natal day 1-3 (P1-P3) pups were injected subcutaneously at the neck area with 20 μl labeling solution (1.5 nmols SNAP-Cell 647-Sir, 2.5 nmols SNAP-Cell Block, 1.5 nmols SNAP-Cell TMR-Star). A skin area of 1.0 × 1.5 cm surrounding the injection site was collected for tissue processing and imaging.

### Isolating basal and luminal mMECs
mMECs was isolated from 11-16 weeks old virgin females. Mammary glands (pairs 2-3 and 4-5) were dissected, finely cut and incubated in 0.01 mg of Collagenase A per 1 g of tissue in mMEC growth media (Advance DMEM/F12 (Life Technologies, 12634028) containing 10% FBS (Gibco, 10270106), 5 ng/ml mEGF (R&D Systems, 2028-WG-200), 5 μg/ml insulin (Sigma-Aldrich, I9278), 1 μg/ml hydrocortisone (Sigma-Aldrich, H4001), 2 mM glutamine (Sigma-Aldrich, 90114 C), 50 μg/ml penicillin and streptomycin) with 10 mM Hepes (Sigma-Aldrich, H3375) shaking at 120 rpm for 2 hours at 37 °C. The cell suspension was centrifuged for 10 min at 400xg to collect the pellet followed by 2-3 pulse centrifugations at 400xg to enrich for mMECs. Next, cell pellets were

treated with 0.05% Trypsin-EDTA (DIFCO, J.T.BAKER) for 7-10 minutes, the cell suspension was filter through a 70 μm cell strainer and centrifuged at 300xg for 5 minutes. The cell pellet was stained with the following antibodies: CD29 FITC (Miltenyi Biotec, 130-102-975), CD326 (Ep-CAM) BV786 (BD; 740958), CD45 PerCP-Cy5.5 (Tonbo Biosciences; 65-0452-U100), CD31 PerCP-Cy5.5 (BD; 562861), Ter-119 PerCP-Cy5.5 (BD; 560512) for 30 minutes on ice (1:500 dilution for all antibodies). After washing with Advance DMEM/F12 base media, cells were resuspended in mMEC growth media with 7AAD (Life Technologies, A1310, 1 μg/ml) for at least 5 minutes on ice before sorting. The basal mMECs were identified with 7AAD⁻, Lin (CD31, CD45, Ter-119)⁻, EpCAM$^{med}$, CD29$^{high}$ and luminal mMECs were identified with 7AAD⁻, Lin (CD31, CD45, Ter-119)⁻, EpCAM$^{high}$, CD29$^{low}$[13,51] using a BD FACSAria Fusion Flow Cytometer (Laser 405 nm, 488 nm, 561 nm, 633 nm) and sorted into mMEC growth media. 7AAD, CD45, CD31, Ter-119 were detected in the Per Cy5.5 channel, EpCAM was detected in the BV786 channel and CD29 was detected in the FITC channel. Single stained samples were used for compensation control.

## mMECs 2D culture

After sorting, mMECs were cultured on fibronectin coated plates (Advanced Biomatrix, 5050, 0.5 μg/ml incubated at 37 °C for 30 minutes) in mMEC growth media with 10 μM Y27632 (BD Biosciences, 562822) overnight followed by mMECs growth media. Experiments were started after removing Y27632 from culture media otherwise specified separately. mMECs were cultured for maximum 7 days without passing. Media was refreshed every 3 days if needed.

## Basal mMECs organoid (3D) culture

After in vivo labeling, isolation and sorting, basal mMECs were labeled with CellTrace Violet (1:1,000 dilution) in MEGM medium (Lonza, CC-3153) according to the manufacturer's instructions. After washing, cells were cultured in mMEC single cell culture media (mMEC growth media supplement with 500 ng/ml R-spondin (R&D Systems, 3474-RS), 100 ng/ml Noggin (PeproTech, 250-38-250ug) on fibronectin coated 12-well plates for 40 hours at 37 °C, 5% CO$_2$ to allow cell division in vitro. Cells were treated with trypsin to detach and collected in mMEC single cell culture media with 7AAD (Life Technologies, A1310, 1 μg/ml) for at least 5 minutes before sorting for 3D culture. Cells that had divided (7AAD⁻, CellTrace Violet$^{low}$) were sub-gated into P$^O$ and P$^Y$ populations based on the amount of old and young peroxisomes.

Cells were sorted into mMECs growth media, centrifuged at 300xg for 5 minutes and plated in 80% Matrigel (Corning, 356231) in mMEC single cell culture media with 10 μM Y27632 for 2 days. For experiments with G6PD inhibitor treatment: G6PDi-1[21] – new G6PD inhibitor with high specific for G6PD (Cayman Chemical Company, 31484) was added during this time to a final concentration of 10 μM. After 2 days, media was changed to mMEC single cell culture media for 6-7 days and refreshed every 2-3 days. For experiments with AGPS inhibitor treatment: AGPS-IN-2i (Aobious, AOB17269) was added during cultured time to a final concentration of 100 μM, media was refreshed every 2-3 days.

For branching assay: following organoid forming assay, media was switched first to mMEC single cell culture media with 2.5 nM FGF2 (Novus Biologicals, NBP2-34921) for 2-3 days and then to mMEC branching media (Advanced DMEM/F12 with 2.5 nM FGF2, 50 μg/ml penicillin and streptomycin, 1X ITS (Sigma-Aldrich, I3146). Media was changed every 2-3 days. Organoids were counted after 5-7 days. For experiments with AGPS inhibitor treatment: AGPS-IN-2i (Aobious, AOB17269) was added during cultured time to a final concentration of 100 μM, media was refreshed every 2-3 days.

## Isolation of EpSCs

EpSCs were isolated from P1-P4 mice. Skin was dissected and incubated in 2 mg/ml dispase II (Roche, 04942078001) in GM2 media

(PromoCell, C-20011), shaking at 120 rpm for 1.5 hours at 37 °C. The epidermis was removed from the dermis and the epidermis was treated with 0.05% Trypsin-EDTA (DIFCO, J.T.BAKER) with 1IU DNAse for 7-10 minutes. Cells were scraped from the epidemis, filtered through a 70 μm cell strainer and centrifuged at 500xg for 5 minutes at +4 °C. The cell pellet was stained with the following antibodies: CD49f FITC (Biolegend, 313605), Sca-1 Brilliant violet 711 (Biolegend, 108131) for 30 minutes on ice (1:500 dilution for all antibodies). After washing with calcium-free DMEM (Gibco, 32068028), cells were resuspended in GM2 media with Sytox blue (Invitrogen, S34857, 2 μM) for at least 5 minutes on ice before sorting. The EpSCs were identified as Sytox blue⁻, CD49$^{high}$ and Sca1$^{high}$. EpSCs were sub-gated into P$^O$ and P$^Y$ populations based on the amount of old and young peroxisomes. Cells were sorted into GM2 media. Sytox blue was detected in the DAPI channel, Sca-1 was detected in the BV711 channel, CD49f was detected in the FITC channel, Snap sir647 was detected in APC channel and Snap TMR star was detected in PE channel. Single stained samples were used for compensation control. BD FACSAria Fusion Flow Cytometer (Lasers 405 nm, 488 nm, 561 nm, 633 nm) was used.

## EpSCs 2D culture

After sorting, EpSCs were cultured on collagen IV coated plates (Bio-Nordika, 5031, 0.5 μg/ml incubated at RT for 2 hours) in GM2 media with 10 μM Y27632 (BD Biosciences, 562822) overnight followed by change into GM2 media. Treatments were started after removing Y27632 from culture media unless specified otherwise. EpSCs were cultured for a maximum 4 days without passing. Media was refreshed every 2-3 days if needed.

## EpSCs organoid (3D) culture

EpSCs were plated in 80% Matrigel (Corning, 356231) in EpSCs organoid culture media (Advance DMEM/F12 (Life Technologies, 12634028), 2 mM glutamine (Sigma-Aldrich, 90114 C), 50 μg/ml penicillin and streptomycin (Sigma-Aldrich, H3375), 50 ng/ml mEGF (R&D Systems, 2028-WG-200), 500 ng/ml R-spondin (R&D Systems, 3474-RS), 100 ng/ml Noggin (PeproTech, 250-38-250ug), 1X B-27 (Fisher Scientific, 11530536), 10 ng/ml Forskolin (Tocris, 1099), 10 μM Y27632 (BD Biosciences, 562822). For experiments with G6PD inhibitor treatment: G6PDi-1[21] (Cayman Chemical Company, 31484) was added for the first 2 days to a final concentration of 10 μM. After 2 days, media was changed to EpSCs organoid culture media for 7-9 days and refreshed every 2-3 days.

## In vitro SNAP labeling of cells in 2D culture

Labeling was done at 37 °C, 5% CO$_2$. Cells were labeled with SNAP-Cell substrates (NEB) for 30 minutes following by three quick washes with PBS and 30 minutes wash in their culture media. All SNAP-Cell substrate stocks were prepared according to manufacturer's instructions, the labeling solution was prepared by diluting the stock solution in cell culture media supplemented with 0.5% BSA to a final concentration of 1.5 μM SNAP-Cell Sir647, SNAP-Cell TMR-star, SNAP-Cell Oregon Green or 4.5 μM SNAP-Cell Block.

## Isolation of hMECs inheriting old or young peroxisomes in ACD

hMECs were synchronized with a double thymidine block[52,53]. Labeling was done at 37 °C, 5% CO$_2$ as described above. In short, cells were treated with 5 mM thymidine (Sigma-Aldrich, T1895) for 19 hours, washed with PBS, detached with 0.05% trypsin-EDTA and collected by centrifugation (400xg for 5 minutes). Cells were treated with SNAP-Cell Block (4.5 μM) for 30 minutes. SNAP-Cell block was then diluted with base DMEM media (Sigma-Aldrich, D7777, at least 10X labeling volume) cells were collected by centrifugation and stained with CellTrace Violet (Thermo Fisher Scientific, C34571) according to manufacturer's instructions. After centrifugation, cells were plated in MEGM. Labeling of old

peroxisomes with SNAP-Cell Sir647 (1.5 μM, 30 minutes, 37 °C, 5% $CO_2$) was performed on dishes 9 hours after releasing from first thymidine followed by a 30-minute wash with MEGM and second thymidine treatment. Cells were released from second thymidine after 17 hours followed by treatment with SNAP-Cell Block (3 μM, 30 minutes) and 30 minutes washes in MEGM. Young peroxisomes were labeled with SNAP-Cell Oregon Green (1.5 μM, 30 minutes) 6 hours after second thymidine release. At 21-22 hours after second thymidine release, cells were detached with 0.05% trypsin-EDTA and collected for cell sorting by centrifugation at 400xg for 5 minutes. Cells that had divided were recognized by low Cell-Trace Violet staining (DAPI channel) and were sub-gated into $P^O$ and $P^Y$ populations based on the amount of old and young peroxisomes (APC and FITC channel). Cells were FACS sorted using a BD FACSAria II Cell sorter (Lasers: Near UV 375 nm, Blue 488 nm, Red 633 nm) or BD FACSAria Fusion Flow Cytometer (see above).

## Mammosphere assay

For mammosphere assays, cells were sorted into culture medium and plated in 1% methyl cellulose (15 cP, Sigma-Aldrich, M7027) in MEGM on 96-well Ultra Low Attachment Microplates (Corning, 3474) at 200-500 cells/ well, 5-10 wells per population. Mammospheres were counted 10-14 days later. For experiments with G6PD inhibitor treatment: G6PDi-1[21] (Cayman Chemical Company, 31484) was added to a final concentration of 30 μM. For experiments with AGPS inhibitor treatment: AGPS-IN-2i (Aobious, AOB17269) was added to a final concentration of 50 and 100 μM.

## Overexpression of G6PD

G6PD overexpression plasmids were generated in a pRP backbone by inserting the G6PD sequence with or without peroxisomal targeting signal (PTS1, YgiM) downstream of the hPGK promoter. In addition, the plasmids contain an mCherry sequence downstream of an IRES sequence. For the control plasmid, the G6PD sequence was replaced by firefly luciferase. mCherry was used as the selection marker for transfected cells in FACS sorting.

To over-express G6PD in hMEC SNAP-PTS1 cells, endotoxin-free plasmids were transfected to the cells using jetPRIME (Polyplus, 101000015) according to the manufacturer's protocol.

## Live cell imaging

Cells were plated on MatTek 35 mm glass bottom dishes, No 1.5 (Mattek, P35G-1.5-14-C). One hour before imaging, cells were stained with Hoechst 33342 (Sigma-Aldrich, 14533, 1 μg/ml) for 12-15 minutes at 37 °C, 5% $CO_2$ followed by a PBS wash. Live cell imaging was performed in cell culture media at 37 °C, 5%$CO_2$ on one of the following platforms:

- 3I Marianas platform (3I intelligent Imaging Innovations) with Zeiss Axio Observer Z1, Optovar 1X, 1.6X and 2.5X, equipped with a Yokogawa CSU-X1 M1 spinning disk, Sutter LB-10W fast 10-position filter wheel with filters: 445/45, 525/30, 617/73, and an Andor Neo sCMOS camera using a 63X Alpha Plan-Apochromat oil objective, NA 1.46 (Carl Zeiss Microscopy). Lasers used were Violet (solid state 405 nm/100 mW), Blue (solid state 488 nm/150 mW), and Lime (solid state 561 nm/50 mW). Images were acquired using Slidebook 5.5 acquisition software (3i).
- Leica SP8 TCS SP8 STED 3X CW 3D platform with Leica DMI8 microscope equipped with HC PL APO 10X/0.40 CS2, 20X/0.75 IMM CORR SC2, 63X/1.20 W motCORR CS2, 93X/1.30 motCORR STED WHITE objectives. Laser used were UV (diode 405 nm/50mw), Blue (Ar 458 nm/7 mW, 476 nm/10 mW, 488 nm/35 mW, 496 nm/10 mW, 514 nm/35 mW), Lime (DPSS 561 nm/20 mW), Red (HeNe 633 nm/12 mW) with beam splitter TD 488/561/633, PMT and HyD GaAsP high sensitivity photon counting detectors. Images were acquired using LAS X software.

## Immunofluorescent staining and imaging of fixed cells and tissues

Cells in 2D culture: Cells were cultured, fixed, stained, and imaged on coverslips (No 1.5) or MatTek 35 mm glass bottom dishes, No 1.5. PBS was used as buffer. Cells were fixed with 4% PFA in PBS for 15 minutes at RT. Cells were permeabilized with 0.1% digitonin or triton-X100 for 10 minutes at RT. Nonspecific binding was blocked with blocking buffer (1%BSA in PBS) for 1 hour at RT. Cells were incubated with primary antibodies in blocking buffer for 1-2 hours at RT. Samples were washed with blocking buffer 3 times, 10-15 minutes each and then incubated with secondary antibodies (1:500 dilution) in blocking buffer for 1 hour at RT. Cells were washed with PBS, and DNA was stained with Hoechst 33342 before imaging. Samples on cover slips were mounted with Immu-Mount (Thermo Scientific, 9990402). Samples on MatTek dishes were imaged in PBS.

Organoids: Organoids were fixed in Matrigel with 4% PFA in PBS for 1 hour at RT. Organoids were scraped from Matrigel and collected in 15 ml Falcon tubes by centrifugation at 300xg for 5 minutes at 4 °C followed by permeabilization with 0.2% triton-X100 in PBS for 30 minutes at RT. Organoids were then incubated with blocking buffer (1%BSA in PBS) for 1 hour at RT. Incubation with primary antibodies was done in 0.05% triton X-100 in blocking buffer overnight at 4 °C under gentle agitation followed by washing with blocking buffer. Incubation with secondary antibodies was done in 0.05% triton X-100 in blocking buffer overnight at 4 °C under gentle agitation. Organoids were washed with PBS, incubated with Hoechst 33342 for 30 minutes and plated in Matrigel on MatTek dishes for imaging.

Skin tissue: SNAP injected animals were sacrificed by cervical dislocation and a minimum of 3 mice per group were used for the experiment and analysis. Dorsal skin was collected and fixed in 4% paraformaldehyde (pH 7.4) at 4 °C overnight. Samples were dehydrated and embedded in paraffin. To avoid duplicates during the analysis and quantification of cell division orientation, 10 micrometer thick sections were used. Tissue sections were deparaffinized and subjected to heat-induced epitope retrieval. Specimens were blocked in 5% goat serum containing 0.2% Triton X-100 and primary antibodies were applied to sections followed by overnight incubation. After washing, sections were incubated with secondary antibodies at RT for 1 hour. Nuclear counterstaining was done by adding DAPI for 10 minutes at RT and the sections were mounted using ProLong Gold (Invitrogen, P36930).

Mammary gland: In vivo SNAP-labeled mammary glands were fixed in 4% PFA in PBS at 4 °C overnight followed by PBS wash. Before imaging, samples were incubated in 80% glycerol supplement with 1 μg/ml Hoechst at RT for 1 hour or at 4 °C overnight for tissue clearance. Mammary glands were placed on coverslip for imaging.

Primary antibodies were mouse monoclonal anti-PMP70 antibody (Sigma-Aldrich, SAB4200181,1:200); rabbit anti-catalase antibody (D4P7B) (Cell Signaling Technology, 12980; 1:600); rabbit polyclonal anti keratin 14 antibody (BioLegend, 905301; 1:500); monoclonal anti-Actin, a-smooth muscle (Sigma-Aldrich, A2547; 1:400); anti-cytokeratin 8 antibody (TROMA-I, DSHB; 1:500), anti a-tubulin antibody (DM1A) (Cell Signaling Technology, 3873, 1:800); anti-Glucose 6 Phosphate Dehydrogenase antibody (Abcam, ab133525; 1:200); anti-E-cadherin (BD, 610182, 1:500), anti-alpha tubulin (Abcam, ab18251, 1:500), anti-Keratin 10 antibody (Biolegend, 905404, 1:300), anti-Keratin 14 antibody (Biologend, 906004).

Secondary antibodies were goat anti rabbit IgG (H + L) Alexa Flour 488 antibody (Invitrogen, A11008), chicken anti-mouse Alexa Flour 488 antibody (Life Technologies, A21200), chicken anti-rat IgG Alexa Flour 488 antibody (Life Technology, A21470), goat anti-mouse Alexa Fluor 488 antibody (Invitrogen, A-11029), Goat anti-rabbit IgG Alexa Fluor 594 antibody (Life Technologies, A11012), Alexa flour donkey anti-mouse 594 antibody (Life Technologies, A21203), Goat anti-rabbit IgG Alexa flour 647 antibody (Life Technologies, A21244); Alexa flour

goat anti-mouse 647 IgG (H + L) antibody (Life Technologies, A21235), goat anti-Rat IgG Alexa Flour 633 (Invitrogen, A-21094).

Images were acquired in one of the below platforms:

- 3I Marianas platform (3I intelligent Imaging Innovations): see above for specs.
- Leica SP8 TCS SP8 STED 3X CW 3D platform: see above for specs.
- Leica TCS SP5 II HCS-A flatform with Leica DMI6000B microscope equipped with HC PL APO 10X/0.40, HC PL APO 20X/0.70, HCX PL APO 20x/0,7 Imm Corr, HCX PL APO 63x/1,2 W Corr/0,17 CS objectives. Laser used was UV (diode 405 nm/50mw), Blue (Ar 458 nm/7 mW, 476 nm/10 mW, 488 nm/35 mW, 496 nm/10 mW, 514 nm/35 mW), Lime (DPSS 561 nm/20 mW), Red (HeNe 633 nm/ 12 mW) with beam splitter QD 405/488/561/633, PMT and HyD detectors. Images were acquired using LAS X software.
- Leica Stellaris 8 Falcon platform with Leica DMI8 microscope equipped with HC PL APO 10X/0.40 CS2, 20X/0.75 IMM CORR SC2, 63X/1.20 W motCORR CS2, 40X/1.25 motCORR. Laser used were UV (diode 405 nm/50mw), white laser Stellaris 8 (440-790 nm). Detectors used were HyD S, HyD X. Images were acquired using LAS X software.
- Zeiss LSM900-Airy imaging systems equipped with EC Plan-Neofluar 5X/0.16, EC Plan-Neofluar 10X/0.3, EC Plan-Neofluar 20X/0.50, Plan-Apo 40X/1.3 Oil, Plan-Apo 63X/1.40 Oil objectives. Laser used were:405 nm, 488 nm, 561 nm, 633 nm with filer set 49 DAPI/ 38 GFP/ 43 DsRed and PMT detectors. Images were acquired using Zen Black v2.1.

## Peroxisome isolation

SNAP labeling of peroxisomes was done in intact hMECs as described above with timeline indicated in the figure. Briefly hMECs were cultured in MEGM media for 48 hours and detached with trypsin-EDTA. After centrifugation, cells were treated with 1.5 μM SNAP-Cell Sir647 for old peroxisomes followed by 30 minutes wash in MEGM. Cells were plated in MEGM for 42 hours followed by SNAP-Cell block (3 μM). Cells were then detached with trypsin-EDTA and collected by centrifugation for young peroxisome labeling with SNAP-Cell TMRstar (1.5 μM). After young peroxisome labeling, cells were treated with 150 nM Mito-Tracker Green FM (Thermo Fisher Scientific, M7514) for 30 minutes. Cells were collected by centrifugation for peroxisome isolation.

Peroxisomes were isolated using Peroxisome Isolation Kit (Sigma-Aldrich, PEROX1) according to the manufacture's instruction. Briefly, cells were homogenized in 1X peroxisome extraction buffer (provided with the kit) by 7 ml Dounce glass tissue grinder (Sigma-Aldrich, T0566) with small clearance pestle (Sigma-Aldrich, P1235), 20-25 strokes. Samples were centrifuged at 1000 x $g$ and then 2000 x $g$ for 10 minutes, and the supernatant was centrifuged at 25,000 x $g$ for 20 minutes. The pellet was collected and suspended in peroxisome extraction buffer using pellet pestle (Sigma-Aldrich, Z35,994-7; Z35,997-1) to obtain a crude peroxisome fraction (CPF). The CPF was mixed with Optiprep Density gradient medium and Optiprep dilution buffer (provided with the kit) to make an Optiprep concentration of 22.5%. The 22.5% mixture was place in between 27.5% and 20% Optiprep solution in a centrifugation tube (Berkman Coulter, 343778) and centrifuged for 1.5 hours at 100,000xg (Optima MAX Ultracentrifuge, TLA 120.2 rotor, Berkman Coulter). The top layers of 20% and 22.5% and the interface at the 22.5%/27.5% was aspirated off. The bottom layer containing the peroxisome fraction was collected and placed on ice for single organelle sorting.

## Single organelle sorting

For separation of different age classes of peroxisomes, the peroxisome fraction was sorted on a BD FACSAria Fusion Flow Cytometer with a 70um nozzle, 1.0xND filer. SNAP-Cell Sir-647, SNAP-Cell TMR Star and MitoTracker Green were detected with APC, PE and FITC channels respectively. APC and PE channels (for detecting old and young peroxisomes) were used for setting the thresholds at 500. The sorting was done with BD FACSDiva software and data analysis was done with the same software or with FlowJo V10.

Peroxisomes were FACS sorted into peroxisome extract buffer supplemented with 1x protease inhibitor (Halt™ Protease Inhibitor, Thermo Scientific, 78446). Sorted peroxisomes were pelleted at 27,000 x $g$ for 25 minutes at 4 °C. The pellets were lysed with RIPA buffer (150 mM NaCl, 20 mM Tris pH7.5, 0.1% SDS, 1% sodium deoxycholate, 1% triton X100) supplemented with protease inhibitor (Halt™ Protease Inhibitor, Thermo Scientific, 78446) and phosphatase inhibitors (PhosSTOP, Merck, 4906845001) and stored at -80 °C for proteomic analysis or western blotting.

## Sample preparation for proteomic analyzes

Sorted peroxisome samples were sonicated for 20 minutes (20 cycles: 30 s ON, 30 s OFF; Bioruptor, Diagenode), and centrifuged at 14,000 x $g$ for 30 minutes at 4 °C and supernatants were collected. Protein concentration was measured by bicinchoninic acid assay (Thermo-Fisher) according to the manufacturer's instructions. Around 10 μg of protein was digested as in previously described protocols[54]. Briefly, proteins were precipitated in cold methanol, incubated at -20 °C for 2 hours, centrifuged at 14,000 x $g$ for 20 minutes at 4 °C, and supernatants were discarded. Protein pellets were re-suspended in 100 mM Tris buffer containing 100 mM dithiothreitol and 4% w/V sodium-dodecyl-sulfate (pH 8.0), heated at 95 °C for 30 minutes under mild agitation, and diluted with 8 M urea in 100 mM Tris buffer (pH 8.0). Samples were loaded on 30 KDa molecular filters (Millipore) and centrifuged at 14,000 x $g$ for 20 minutes. Filters were washed twice with 8 M urea buffer, incubated in 50 mM iodoacetamide in 8 M urea buffer for 30 minutes (in the dark), and washed 4 times (2x 8 M urea buffer, 2x 50 mM tri-ethyl-ammonium bicarbonate buffer pH 8.0). Proteins were then digested using trypsin (enzyme-protein ratio 1:50) at 37 °C for 16 hours at 650RPM. After digestion, filters were centrifuged at 14,000 x $g$ for 20 minutes to extract tryptic peptides.

Peptide mixtures were desalted and cleared of residual contaminants by sequential solid phase extraction. First, samples were loaded on primed (priming wash sequence: methanol, 80% acetonitrile (ACN) and 0.5% acetic acid in water, 0.5% acetic acid in water) C18 tips (Sigma), washed with 100 μl of 0.5% acetic acid in ultrapure water, and eluted in 0.5% acetic acid and 80% ACN in ultrapure water. Eluates were dried in a vacuum concentrator. Second, eluates were dried and subjected to SP3 peptide purification[55]. A volume of 2 μl of SP3 beads (1:1 ratio of Sera Mag A and Sera Mag B re-suspended in ultrapure water; Sigma) was added to dried peptides and incubated for 2 minutes under gentle agitation. A volume of 200 μl of ACN was then added and samples were incubated for 10 minutes. This step was repeated twice. Peptides were eluted from beads by adding 200 μl of 2% dimethyl sulfoxide in water and sonicating mixtures for 1 minute. Supernatants were then collected, dried, and stored at -80 °C until MS analysis.

## Proteomics analysis

MS analysis was performed on a Q-Exactive HF-X (Thermo-Fisher) mass spectrometer. Around 1 μg of tryptic peptides was separated on a RP-HPLC EasySpray column (ID 75 μm × 25 cm C18 2 μm 100 Å resin; Thermo-Fisher) coupled to a nano-LC EASY 1200 chromatography system (Thermo-Fisher). Peptides were first trapped on an Acclaim PepMap™ 100 pre-column (ID 75 μm × 2 cm C18 3 μm, 100 Å resin; Thermo-Fisher), then separated on a reverse phase HPLC EasySpray column (ID 75 μm × 50 cm C18 2 μm, 100 Å resin; Thermo-Fisher). Chromatographic gradient was run at a flow of 350 nl/min with the following steps: 10-30% B in 90 minutes; 30-45%B in 20 minutes; 45-95% B in 1 minute; 95%B for 9.5 minutes. Full MS scan parameters: 60,000 resolution, AGC target was set to 3E6. MS/MS scan parameters: 15,000 resolution, AGC target was set to 1E5. Collision energy was set to 28, n of analyzed peaks was set to 15. Dynamic exclusion window was set to 10 s.

## Proteomics data processing

MS analysis-derived RAW files were searched using the Andromeda[56] search engine as part of the MaxQuant[57] software suite (v1.6.14.0). Spectra were searched against the Uniprot-Swissprot human proteome database (version download: 2020.02.24). Trypsin was selected as protease for in silico digestion. Carbamidomethylation of Cysteine residues was selected as fixed modification, while Methionine oxidation and acetylation of N-terminal residues were selected as variable ones. Label-free Quantification (LFQ)[58] was activated to allow accurate evaluation of protein abundance. Identification of peptides resulting from missed cleavages was allowed. Precursor ion tolerance was set to 20 and 4.5 ppm for first and main searches, respectively. Match-between-runs option was enabled, and settings left to default. The protein intensity table resulting from the search was filtered for protein q-value (<0.01), contaminant (excluded), reverse sequences (excluded), and unique peptides (at least 1). Protein table was imported in R (v3.6), filtered for missing values (at least 50% observations in one sample group) and $Log_2$ transformed. Residual missing observations were imputed with 0 prior centering and scaling the dataset.

## Western blotting

Cell or peroxisome samples were lysed with RIPA buffer supplemented with protease and phosphatase inhibitors (as above). Samples were vortexed or sonicated (5 cycles: 30 s on, 30 s off, at 4 °C in a Bioruptor sonicator) and centrifuged at 13,000xg for 20 minutes and the supernatant was collected. Protein concentrations were measured using Bio-Rad Protein Assay (Bio-Rad, 5000001) or Pierce BCA Protein assay kit (Thermo Fisher Scientific, 23225).

Western blots were run with the Bolt system (Thermo Fisher Scientific). Proteins were denatured by adding Bolt LDS sample buffer (B0007) and sample reducing agent (B0009), heating to 95 °C for 3-5 minutes followed by placing samples on ice immediately. Samples were run on Bolt 4-12% Bis-Tris Gels (NW04120BOX) in Bolt MES SDS running buffer (B0002) and resolved proteins were transferred to 0.2 μM nitrocellulose membrane with Bolt transfer buffer (BT00061) according to the manufacturer's instruction.

Membranes were incubated in blocking buffer (5% skim milk, 0.1% Tween-20 in TBS) for 1 hour at RT followed by incubation with primary antibodies in blocking buffer overnight at 4 °C. Membranes were washed with blocking buffer, incubated with HRP conjugated secondary antibodies in blocking buffer for 1 hour at RT and washed with 0.1% Tween-20 in TBS. Proteins were detected using Pierce ECL Western Blotting Substrate (Thermo Fisher Scientific, 10005943) or SuperSignal West Femto Maximum Sensitivity Substrate (Thermo Scientific, 34094) and Fujifilm Super RX films (FUJRX18 × 24) or ChemiDoc MP (Bio-Rad).

Primary antibodies were mouse monoclonal anti-PMP70 antibody (Sigma-Aldrich, SAB4200181,1:800); rabbit anti-catalase antibody (D4P7B) (Cell Signaling Technology, 12980; 1:500); mouse a-tubulin (DM1A) antibody (Cell Signaling Technology, 3873, 1:1000); anti-SNAP-tag antibody (New England BioLabs, P9310S; 1:1,000); anti-beta Actin antibody (Abcam, ab8227, 1:2,000); Anti-UQCRFS1 [EPR16288] (Abcam, ab191078, rabbit, 1:1,000); anti-LAMP1 antibody (Abcam, ab24170, 1:1,000); Calnexin polyclonal antibody (Enzo, ADI-SPA-860, 1:1,000), anti-Glucose 6 Phosphate Dehydrogenase antibody (Abcam, ab133525, 1:200).

Secondary antibodies were anti-mouse IgG HRP-linked antibody (Cell Signaling Technology, 7076S, 1:1,000) or anti-rabbit IgG HRP-linked antibody (Sigma-Aldrich, A0545, 1:1000-3000).

Analyze Gels function of Fiji ImageJ software[59] was used for Western blots quantification. Protein level was calculated against housekeeping proteins α-tubulin or β-actin.

## Metabolomics

50000 hMECs were sorted using FACS and resuspended in assay medium (glucose-, pyruvate- and glutamine-free DMEM (Agilent, 103334-100) supplemented with MEGM supplements (Lonza, CC-3150) and 8 mM 1,2-13 C glucose (Cambridge Isotope Laboratories, CLM-504-0.25). Cells were immediately seeded on 48 well plates in a total volume of 200 μl. Samples of 5 μl medium were collected 0 and 1 hour later in 95 μl ice cold 80% acetonitrile and stored at -80C until analysis.

All samples were analyzed on a Thermo Q Exactive Focus Quadrupole Orbitrap mass spectrometer coupled with a Thermo Dionex UltiMate 3000 HPLC system (Thermo Fisher Scientific, Inc.). The HPLC was equipped with a hydrophilic ZIC-pHILIC column (150 × 2.1 mm, 5 μm) with a ZIC-pHILIC guard column (20 × 2.1 mm, 5 μm, Merck Sequant). 5ul of the samples were injected into the LC-MS after quality controls in randomized order having every 10th samples as blank. The separation was achieved by applying a linear solvent gradient in decreasing organic solvent (80–35%, 16 min) at 0.15 ml/min flow rate and 45 °C column oven temperature. The mobile phases were following, aqueous 200 mmol/l ammonium bicarbonate solution (pH 9.3, adjusted with 25% ammonium hydroxide), 100% acetonitrile and 100% water. The amount of the ammonium bicarbonate solution was kept at 10% throughout the run resulting in steady 20 mmol/l concentration. Metabolites were analyzed using a MS equipped with a heated electrospray ionization (H-ESI) source using polarity switching and following setting: resolution of 70,000 at m/z of 200, the spray voltages: 3400 V for positive and 3000 V for negative mode, the sheath gas: 28 arbitrary units (AU), and the auxiliary gas: 8 AU, the temperature of the vaporizer: 280 °C, temperature of the ion transfer tube: 300 °C. Instrument control was conducted with the Xcalibur 4.1.31.9 software (Thermo Scientific). The peaks for metabolites were confirmed using commercial standards (Sigma etc). Data quality was monitored throughout the run using inhouse quality control cell line extracted in a same way as unknown samples and blank as every 10th sample. The final peak integration was done with the TraceFinder 4.1 SP2 software (Thermo Scientific) and for further data analysis, the peak area data was exported as an excel file. The ratio of M + 1 to M + 2 lactate was calculated from the peak areas.

## Lipidomics

Lipids were extracted from cell pellets (at least from one million cells) according to Folch et al. [60] and dissolved in chloroform/methanol 1:2 (v/v). Internal standards [EquiSPLASH® internal standard mixture and Ceramide (Cer) 18:1;O2/17:0, both from Merck] were added and samples were analyzed with LC-MS/MS as described before[61]. The method used acetonitrile/water/isopropanol-based solvent system[62], Agilent 1290 Infinity HPLC (Agilent Technologies, Santa Clara, CA) equipped with a Luna Omega C18 100 Å (50 × 2.1 mm, 1.6 μm) column (Phenomenex) and Agilent 6490 Triple Quad LC/MS with iFunnel Technology. The lipids species were identified and quantified using lipid class-specific precursor ion and neutral loss detection modes, as previously described[61]. Additionally, PE plasmalogens (PE P) were identified with alkenyl chain (16:0, 18:0, 18:1)-specific scans as described previously[63] and quantified from MS- scan. Mass spectra were processed using MassHunter Qualitative Navigator software (Agilent) and lipid species were quantified utilizing the internal and additional external standards (glucosylceramide 18:1;O2/24:1) and LIMSA software[64]. Lipid data are expressed as molar percentages (mol%) and lipid species are marked as follows: [sum of acyl chain carbons]: [sum of acyl chain double bonds] (e.g., PC 36:1).

## Analysis of fatty acyls, alkenyls and alcohols

Fatty acyl (FA) and alkenyl chains, and fatty alcohols (FOHs) were analyzed by gas chromatography (GC) following previously published procedures[65–67]. One half of the Folch-extract of the cell pellet was evaporated near to dryness under nitrogen flow and the extracted lipids were transmethylated by heating with 1% $H_2SO_4$ in methanol under nitrogen atmosphere, which converts FAs to their methyl esters, phospholipid-derived alkenyl chains to dimethyl acetals (DMAs) and leaves FOHs intact. After adding water, the FAs, DMAs and FOHs were

extracted with hexane, and the sample solution was dried, and concentrated. These samples were then injected into a GC-2010 Plus gas chromatograph (Shimadzu Scientific Instruments, Kyoto, Japan) with flame-ionization detector for quantification of the analytes, and their identification was performed using a GCMS-QP2010 Ultra (Shimadzu Scientific Instruments, Kyoto, Japan) with mass selective detector (MSD). Both systems were equipped with Zebron ZB-wax capillary columns (30 m, 0.25 mm ID and film thickness 0.25 μm; Phenomenex, Torrence CA, USA). The compositions of the lipid-derived FAs and DMAs were expressed as mol% profiles, and the FOHs were calculated as μmol/10$^6$ cells.

### Image analysis

Asymmetric inheritance of peroxisomes in division pairs was quantified using Fiji ImageJ software. All the stacks were z-projected with sum intensity. Cell borders were identified using bright filed (cells in 2D culture) or E-cadherin/ α-tubulin (EpSC). Total intensity of old or young peroxisomes in two daughter cells was considered 100%.

For colocalization analysis of G6PD with peroxisomes Imaris x64 software was used. Peroxisomes were identified using the Spots creation function based on both old and young peroxisome labeling. Old and young label intensity of each spot was normalized with the median value of each channel and the old/young ratio was calculated based on the normalized value. Total peroxisome number of each cell was divided equally into 3 different groups: old, double positive, and young peroxisomes based on old/young ratio ranking: i.e., one third of peroxisomes with highest old/young ratio was classified as old peroxisomes, one third of peroxisomes with lowest old/young ratio was classified as young peroxisomes, and the rest was double positive peroxisomes. G6PD was identified using the spot creation function. Colocalization of G6PD and peroxisomes was analyzed using the Colocalize spots function (extension of Imaris x64 software) with the distance between center of 2 spots set at 0.5 μm. Colocalized peroxisomes were classified into different age groups of peroxisomes based on their old/young ratio and their fraction of colocalization with G6PD was quantified.

### Statistic and reproducibility

Two-tailed Student's t-test or one-way ANOVA were used to compare groups otherwise indicated separately, and $p < 0.05$ was considered significant. The statistical analysis of RNA sequencing and mass spectrophotometry is described above. Microsoft Excel (Office 16) or GraphPad Prism 8 were used for statistical analysis.

No statistical method was used to predetermine the sample size. No data were excluded from the analyses. The investigators were not blinded to allocation during experiments and outcome assessment.

### Reporting summary

Further information on research design is available in the Nature Portfolio Reporting Summary linked to this article.

## Data availability

The mass spectrometry proteomics data has been deposited to the ProteomeXchange Consortium via the PRIDE partner repository[68] with the dataset identifier PDX028679 [https://www.ebi.ac.uk/pride/archive/projects/PXD028679]. The metabolomics data has been deposited to MetaboLights[69] repository with the study identifier MTBLS12309. The lipidomics data has been deposited to Metabolomics Workbench[70], with the study ID ST003808 [https://doi.org/10.21228/M8QZ7Q]. Source data are provided with this paper.

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

## Acknowledgements
The authors thank Jenny Bärlund, Maija Simula and Ella Kärkkäinen for technical assistance, all the Katajisto laboratory's members for discussion and comments. This study was carried out with the support of HiLIFE and its infrastructures: HiLIFE Light Microscopy Unit at the Institute of Biotechnology; HiLIFE Laboratory Animal Center Core Facility; and by Karolinska Institutet: Microscopy unit at Biomedicum. This study was funded by grants from European Research Council ERC #677809 and #101045009 (P.K.), Academy of Finland #266869, #304591, #312436, #320185, #336194, #353000 (P.K.), Chan Zuckerberg Initiative MET-0000000418 (P.K.), Knut and Alice Wallenberg Foundation KAW 2014.0207 and 2022.0054 (P.K.), Swedish Research Council 2018-03078, 2018-02963, 2022-01304 (P.K.), Cancerfonden 190634, 180681, and 222499 (P.K.), Center for Innovative Medicine CIMED (P.K.), Sigrid Juselius Foundation (P.K.), Cancer Society of Finland (P.K.), Doctoral Programme in Biomedicine at the University of Helsinki (H.B.), the Finnish Cultural Foundation (H.B.), Maud Kuistila Memorial Foundation (H.B.).

## Author contributions
P.K. and H.B. conceived the study. H.B., S.A., A.S.C. and P.K designed and performed the ex vivo and in vivo experiments and interpreted the results. H.B., S.A., E.V., E.K., B.R., V.H. and P.K. designed and performed the in vitro experiments and interpreted the results. T.D.M. and E.N. performed mass spectrometry analysis and analyzed proteomics results. M.H., A.H and R.K. performed lipidomic profiling and analyzed the results together with H.B. The manuscript was written by H.B. and P.K. with inputs from S.A., E.K., A.S.C., V.H., J.I.E., M.K. and all co-authors.

## Competing interests
The authors declare no competing interests.
