## [Transparent Peer Review file · Nature Communications]

Glucose-6-phosphate-dehydrogenase on old peroxisomes maintains self-renewal of epithelial stem cells after asymmetric cell division

Corresponding Author: Professor Pekka Katajisto

Version 0:

Reviewer comments:

Reviewer #4

(Remarks to the Author)

The manuscript "G6PD on old peroxisomes maintains self-renewal of epithelial stem cells after asymmetric cell division" by Bui and colleagues describes the asymmetric apportioning of old and young peroxisomes (O-P and Y-P) in murine mammary and skin stem cells. The authors show that O-Ps carry more Glucose-6-phosphate-dehydrogenase (G6PD) enzymes, and this acts as determinant for the stem cell fate probed by organoid forming assays. The finding that O-P and Y-P are asymmetrically segregated in mammary and skin stem cells is novel and interesting for scientists in the stem cell field. Unfortunately the molecular mechanism whereby O-P and Y-P are asymmetrically inherited remains uncharacterised.

Reviewer #5

(Remarks to the Author)

The authors have revised their manuscript appropriately have addressed my comments and the comments of the other reviewers. This work will be well cited and is an important advance in the stem cell and organelle biology fields.

Reviewer #6

(Remarks to the Author)

The asymmetric segregation of peroxisomes in daughter cells of dividing mammalian stem cells is a very novel addition to the knowledge of peroxisome dynamics. Cutting edge techniques were developed to convincingly proof that old peroxisomes are preferentially inherited by the cells that retain stemness.

By again using novel technologies, the authors further claim that old peroxisomes distinguish from young peroxisomes because they contain G6PD. This would be important for the generation of NADPH at the peroxisomal membrane which is essential for two reactions in ether phospholipid synthesis. The latter lipids would determine cellular fate.

According to the proteomics analysis there is indeed a clear difference in expression of G6PD in old vs young peroxisomes. However, the way this is described in results and discussion as well as some of the data require a much more critical assessment and more accurate formulation.

"...in line with earlier works (ref 20,21), we found that a subset of G6PD indeed associates with peroxisomes". It should be made clear (already in the results) that G6PD is an abundant cellular protein that resides primarily in the cytoplasm, and was occasionally detected in peroxisomes in two old studies. In a major inventory of the mammalian peroxisomal proteome made by leaders in the field (Yifrach et al, Defining the Mammalian Peroxisomal Proteome, 2018) G6PD received a very low score, so it should be emphasized that this is not a bonafide peroxisomal protein. An important obstacle here is that the targeting mechanism of this protein to the peroxisome remains obscure.

Further, the authors overexpress G6PD in three locations (cytosol, peroxisomal matrix and cytoplasmic side of peroxisomal membrane). They state that pmembG6PD mimics the localization on old peroxisomes. Although I might overlook this, I do

not find how the authors prove that the enzyme is located on the outside of the membrane of old peroxisomes? Which figure? In ref 20,21, it is rather assumed that G6PD is 'in' peroxisomes.

Line 169: "Interestingly, peroxisomal ether lipid synthesis depends on two NADPH-dependent enzymes residing on the peroxisomal membrane FAR1/2 and ADHAPR (renamed PEXrap) (scheme in Fig2d)". However, Honsho et al. (Front Cell Dev Biol 8: 855, 2020) showed that although ADHAPR is present on the peroxisomal membrane, it is the ER version of this enzyme that does the bulk of the reduction of DHAP to G3P. This is now generally accepted in the field (Wanders et al, Physiol Rev, 2023, Figure 3). Also the sentence in line 180-181 is not accurate. This needs to be corrected.

To show that G6PD on the peroxisomal membrane promotes ether lipid synthesis, lipidomes of cG6PD and pmembG6PD were compared. I am not convinced by the outcome and am skeptical about the conclusions that are drawn. i) As stearic acid has many different fates in a cell, I doubt that an (moderate) increased usage in ether lipid synthesis (more reduction by FAR1/2 to the alcohol) would result in a decrease of the total stearic acid that is stored (a transmethylation was performed). ii) the trend towards an increase of the FAR1/2 product octadecanol solely depends on 1 datapoint that is much higher in the pmembG6PD condition than the 7 other datapoints (from both conditions) iii) there is indeed a slight increase in plasmalogen content (less than 10%, albeit significant). It is for me hard to imagine how this small change can be determinant for the fate of the daughter cell.

In the discussion, it is stated that further research on the way G6PD is recruited to (old) peroxisomes "could provide valuable insights". This is an understatement because the validity of the current data strongly depends on understanding how the protein can be preferentially targeted to old peroxisomes.

Version 1:

Reviewer comments:

Reviewer #6

(Remarks to the Author)

The authors adequately improved the text by more accurate background, descriptions and interpretation of the data. One issue remains i.e. the lipidomics analysis in extended fig 6b.

I agree that quantifying fatty alcohols might be very demanding. However, claiming that there is a trend to an increase, while this depends on a single data point that is 3 – 4 times higher than all the other data points, is not scientifically correct in my opinion. This is misleading. The authors did not respond to my previous comment.

For the increase in ether lipid levels, the authors should clarify in the text that the 'significant' increase is x% . Significance and relevance does not always coincide.

Please find our responses to the Reviewer comments below. Our responses are in blue, and changes to the manuscript text are indicated in red, with a strikethrough indicating deleted parts.

Reviewer #4 (Remarks to the Author):

The manuscript "G6PD on old peroxisomes maintains self-renewal of epithelial stem cells after asymmetric cell division" by Bui and colleagues describes the asymmetric apportioning of old and young peroxisomes (O-P and Y-P) in murine mammary and skin stem cells. The authors show that O-Ps carry more Glucose-6-phosphate-dehydrogenase (G6PD) enzymes, and this acts as determinant for the stem cell fate probed by organoid forming assays.

The finding that O-P and Y-P are asymmetrically segregated in mammary and skin stem cells is novel and interesting for scientists in the stem cell field. Unfortunately the molecular mechanism whereby O-P and Y-P are asymmetrically inherited remains uncharacterised.

We thank the reviewer for her/his positive feedback on the significance of our findings. We fully agree that the mechanistic insights on the sorting mechanism (underpinning the age-selective segregation) would be of very high value and allow targeting of the mechanism in downstream work. During the lengthy project we have managed to exclude multiple candidates, but decided to not include that work based on the feedback from the reviewers. This was done to emphasize the discovery of the metabolic consequences resulting from the yet to illuminate sorting mechanism.

Reviewer #5 (Remarks to the Author):

The authors have revised their manuscript appropriately have addressed my comments and the comments of the other reviewers. This work will be well cited and is an important advance in the stem cell and organelle biology fields.

We thank the reviewer for appreciating our responses and the revision work. We are particularly thankful for the recognition of the importance of our findings.

Reviewer #6 (Remarks to the Author):

The asymmetric segregation of peroxisomes in daughter cells of dividing mammalian stem cells is a very novel addition to the knowledge of peroxisome dynamics. Cutting edge techniques were developed to convincingly proof that old peroxisomes are preferentially inherited by the cells that retain stemness.

By again using novel technologies, the authors further claim that old peroxisomes distinguish from young peroxisomes because they contain G6PD. This would be important for the generation of NADPH at the peroxisomal membrane which is essential for two reactions in ether phospholipid synthesis. The latter lipids would determine cellular fate.

According to the proteomics analysis there is indeed a clear difference in expression of G6PD in old vs young peroxisomes. However, the way this is described in results and discussion as well as some of the data require a much more critical assessment and more accurate formulation.

"...in line with earlier works (ref 20,21), we found that a subset of G6PD indeed associates with peroxisomes". It should be made clear (already in the results) that G6PD is an abundant cellular protein that resides primarily in the cytoplasm, and was occasionally detected in peroxisomes in two old studies. In a major inventory of the mammalian peroxisomal proteome made by leaders in the field (Yifrach et al, Defining the Mammalian Peroxisomal Proteome, 2018) G6PD received a very low

score, so it should be emphasized that this is not a bonafide peroxisomal protein. An important obstacle here is that the targeting mechanism of this protein to the peroxisome remains obscure.

We want to start by thanking the Reviewer for recognizing the impact of our discoveries on age-selective peroxisome segregation and how it influences cell fate.

Regarding the description of our findings on G6PD, we fully agree with the reviewer and have modified the text accordingly (including the reference mentioned):

-Revised manuscript line 137-138: “Although G6PD is primarily localized in cytoplasm, in line with some earlier works ^{19,20,21}, we found that a subset of the abundant G6PD indeed associates with peroxisomes (Fig.2b).”

- Revised manuscript line 230-231: “While we cannot exclude the existence of additional mechanisms, we here discover the novel role of typically cytoplasmic G6PD specifically on peroxisomal metabolism and how it regulates cell fate through ether lipid biosynthesis.”

Further, the authors overexpress G6PD in three locations (cytosol, peroxisomal matrix and cytoplasmic side of peroxisomal membrane). They state that pmembG6PD mimics the localization on old peroxisomes. Although I might overlook this, I do not find how the authors prove that the enzyme is located on the outside of the membrane of old peroxisomes? Which figure? In ref 20,21, it is rather assumed that G6PD is 'in' peroxisomes.

This is a relevant point, and we agree that it warrants clarification.

We targeted G6PD to peroxisomes with two different targeting signals (matrix and membrane) for the following reasons:

- 1. Our immunofluorescent staining for G6PD (Extended data Fig5c) suggested its localization might actually reside on the peroxisomal membrane. However, we agree that the IF imaging does not provide conclusive data on the matter.*
- 2. Studies by Patel et al (1987), and Panchenko & Antonenkov (1984) suggest G6PD is localized in the peroxisomal matrix. Notably, these studies did not visually demonstrate G6PD's presence on the peroxisomal membrane but inferred it indirectly by collected peroxisomal matrix proteins after disrupting the peroxisomal membrane. This approach, however, does not rule out the possibility that the use of KCl may have destabilized the peroxisomal membrane (Panchenko et al, 1976), potentially releasing proteins weakly interacted with the peroxisomal membrane into the matrix fraction.*
- 3. Electron microscopy data from Frederiks & Vreeling-Sindelarova (2001) visually demonstrated G6PD activity predominantly at the periphery of peroxisomes (Figure 3), suggesting that its activity mainly resides at the peroxisomal membrane.*

Figure3 in Frederiks & Vreeling-Sindelarova (2001)

As pointed out by the Reviewer, G6PD is primarily cytoplasmic, making association with the peroxisomal membrane on the cytoplasmic side plausible. Moreover, targeting G6PD to the peroxisomal membrane (Fig. 2c) and replicated the phenotype separating cells that inherit old peroxisomes from those inheriting younger peroxisomes (Fig. 1e).

Taken together, the evidence on peroxisomal localization of G6PD by previous reports is not consistent, and our data indicates a role specifically for membrane associated G6PD on the cytoplasmic side. However, as noted by the Reviewer, we do not formally demonstrate the localization of old peroxisomes on the cytoplasmic face of peroxisomal membrane, and have therefore revised the text accordingly.

Revised line 151: “Excitingly, p^{memb} G6PD ~~that mimics the localization of G6PD on old peroxisomes~~, increased the mammosphere formation, whereas targeting to the other two locations had no impact (Fig.2c, Extended data Fig.5e,f).”

Line 169: "Interestingly, peroxisomal ether lipid synthesis depends on two NADPH-dependent enzymes residing on the peroxisomal membrane FAR1/2 and ADHAPR(renamed PEXrap) (scheme in Fig2d)". However, Honsho et al. (Front Cell Dev Biol 8: 855, 2020) showed that although ADHAPR is present on the peroxisomal membrane, it is the ER version of this enzyme that does the bulk of the reduction of DHAP to G3P. This is now generally accepted in the field (Wanders et al, Physiol Rev, 2023, Figure 3). Also the sentence in line 180-181 is not accurate. This needs to be corrected.

We thank the reviewer for these important insights. We are aware of the ER localization of ADHAPR, but as it also resides on peroxisomal membrane, we assumed that its likelihood of utilizing NADPH produced by G6PD would be similar to that of FAR1/2. However, based on reviewer’s expert feedback, we have now revised Fig2d and the text accordingly (including the references mentioned).

Revised line 170-172: Interestingly, peroxisomal ether lipid synthesis depends **primarily** on two NADPH-dependent enzymes residing on the peroxisomal membrane, namely Fatty acyl-CoA reductases **1 and 2 (FAR1 and FAR2)**²⁸⁻³¹ ~~and Acyl/alkyl DHAP reductase (ADHAPR)~~ (Fig. 2d).

Revised line 181-183 (formal manuscript 180-181) “To address if the increased peroxisomal ether lipid synthesis contributes to stemness maintenance, we used AGPS-IN-2i to inhibit Alkylglycerone-Phosphate Synthase (AGPS). **AGPS acts in the peroxisomal matrix, converting fatty alcohol - produced by FAR1/2 activity on the peroxisomal membrane - into alkyl-DHAP and couples the NADPH-consuming reactions of FAR1/2 and ADHAPR at the membrane** (Fig. 2d).”

Revised Fig.2d

To show that G6PD on the peroxisomal membrane promotes ether lipid synthesis, lipidomes of cG6PD and pmembG6PD were compared. I am not convinced by the outcome and am skeptical about the conclusions that are drawn. i) As stearic acid has many different fates in a cell, I doubt that an (moderate) increased usage in ether lipid synthesis (more reduction by FAR1/2 to the alcohol) would result in a decrease of the total stearic acid that is stored (a transmethylation was performed). ii) the trend towards an increase of the FAR1/2 product octadecanol solely depends on 1 datapoint that is much higher in the pmembG6PD condition than the 7 other datapoints (from both conditions) iii) there is indeed a slight increase in plasmalogen content (less than 10%, albeit significant). It is for me hard to imagine how this small change can be determinant for the fate of the daughter cell.

These are excellent points, and as we had no means to directly address the ether lipid synthesis rates, we had to rely on insights from lipidomics on total amounts. This has the potential to mask differences in activity as the cells are actively dividing, and contribute to the relatively modest differences noted in our data. Specific points i-iii) by the reviewer:

- i) *This is a valid point, but as plasmalogen PE P-36:1 is primarily composed of 18:0p/18:0, and two FA 18:0 molecules are used for each molecule of PE P-36:1, the increase in ether lipid synthesis impacts the FA 18:0 pool with a factor of 2. While the relatively large stearic acid pool is indeed used widely in the cell, we note a change that is consistent with increase in ether lipid synthesis, but arguably we cannot conclude the effect is only due to increase in ether lipid synthesis, and neither do we claim so.*
- ii) *Lipid synthesis operates through dynamically regulated pools. When fatty acids (FAs) are converted to fatty alcohols (FOH), the extra FOH that is not utilized for ether lipid or other synthesis is likely converted back to FAs (Rizzo 2014, Figure 4)¹. Therefore, a substantial accumulation of FOH or plasmalogens is unlikely unless all regulatory mechanisms are overwhelmed. While possible, this is not commonly observed in living, actively growing cells, especially when only G6PD is overexpressed, as it provides NADPH – a coenzyme for FAR1/2. Additionally, detecting fatty alcohol is very challenging due to: 1) their low abundance in biological extract, 2) poor ionization efficiency, 3) higher volatility compared to FA during sample preparation, and 4) the need for derivatization (Cao et al., 2014)². Only two fatty alcohols were identified in our lipidomic (FOH 16:0 and FOH 18:0). However, we included the data to illustrate the trends in the multiple steps of the ether lipid synthesis pathway. Importantly, we did not claim the data to show a significant change in octadecanol.*
- iii) *We understand the reviewer's skepticism regarding the correlation between small changes in plasmalogens and cell fate. However, even small changes in membrane glycerophospholipid and FAs have been shown to have significant impact on the fate of mesenchymal stem cell when cultured in vitro (Chatgialloglu et al, 2017)³, and to influence preimplantation mammalian embryo development (Zhang et al, 2024)⁴.*

Moreover, the changes observed in our lipidomic analysis should be considered in the light of the chosen experimental strategy. We overexpressed cG6PD and p^{mem}G6PD transiently, and lipidomic samples were collected two days post-transfection. As the production of this data was additional to the already heavy revision work we conducted based on the reviewer comments, we did not collect samples from a time series experiment that would have allowed identification of the timepoint with most robust induction in lipidomic alterations.

Finally, we wish to highlight that in addition to the lipidomic analysis, which guided our hypothesis on the role of G6PD at the peroxisomal membrane, we specifically inhibited AGPS in the G6PD over-expressing cells (Fig.2e). This inhibition effectively abolished the enhancement of stemness induced by p^{mem}G6PD. Taken together, our findings indicate that G6PD on peroxisomes contributes to fate determination. We also demonstrate in two stem cell

systems (skin and mammary) that inhibition of AGPS induces a reduction in stem cell activity that is similar to that induced by inheritance of young peroxisomes. Finally, we show in one stem cell system (mammary) that the phenotypes associated with peroxisomal G6PD (either age-selective inheritance or p^{memb} G6PD) are mitigated by the AGPS inhibition, and show lipidomic changes that are in agreement of ether lipid synthesis being influenced by the peroxisomal G6PD. In no point of the manuscript do we indicate that changes in the ether lipid synthesis would be the sole alteration induced by age-selective inheritance of old peroxisomes. However, in response to Reviewer points, we have now modified the discussion to clearly state this point.

Revised line 230-231: "While we cannot exclude the existence of additional mechanisms, we here discover the novel role of typically cytoplasmic G6PD specifically on peroxisomal metabolism and how it regulates cell fate through ether lipid biosynthesis."

In the discussion, it is stated that further research on the way G6PD is recruited to (old) peroxisomes "could provide valuable insights". This is an understatement because the validity of the current data strongly depends on understanding how the protein can be preferentially targeted to old peroxisomes.

We have edited the text accordingly.

Revised line 239-240: "Consequently, further studies on how G6PD is recruited to peroxisomes, especially older ones, and how peroxisomal ether lipid synthesis affects stem cell functions are crucial for advancing our understanding of these processes."

References

1. Rizzo, W. B. Fatty aldehyde and fatty alcohol metabolism: Review and importance for epidermal structure and function. *Biochimica et Biophysica Acta - Molecular and Cell Biology of Lipids* vol. 1841 377–389 Preprint at <https://doi.org/10.1016/j.bbalip.2013.09.001> (2014).
2. Cao, Y. *et al.* N-alkylpyridinium quaternization combined with liquid chromatography-electrospray ionization-tandem mass spectrometry: A highly sensitive method to quantify fatty alcohols in thyroid tissues. *Anal Chim Acta* 849, 19–26 (2014).
3. Chatgillaloglu, A. *et al.* Restored in vivo-like membrane lipidomics positively influence in vitro features of cultured mesenchymal stromal/stem cells derived from human placenta. *Stem Cell Res Ther* 8, 1–11 (2017).
4. Zhang, L. *et al.* Low-input lipidomics reveals lipid metabolism remodelling during early mammalian embryo development. *Nat Cell Biol* 26, 278–293 (2024).

Please find our responses to the Reviewer comments below. The reviewer's comments are in black, and our responses are in blue.

Reviewer #6 (Remarks to the Author):

The authors adequately improved the text by more accurate background, descriptions and interpretation of the data.

We thank the reviewer for appreciating our responses and the revision of the manuscript.

One issue remains i.e. the lipidomics analysis in extended fig 6b.

I agree that quantifying fatty alcohols might be very demanding. However, claiming that there is a trend to an increase, while this depends on a single data point that is 3 – 4 times higher than all the other data points, is not scientifically correct in my opinion. This is misleading. The authors did not respond to my previous comment.

We have now removed the sentence “Moreover, pmembG6PD cells showed a trend towards increase in the FAR1/2 product 1-octadecanol (FOH 18:0)”.

For the increase in ether lipid levels, the authors should clarify in the text that the ‘significant’ increase is x% . Significance and relevance does not always coincide.

We agree with the reviewer, and have modified the text accordingly:

Line 173-176: “While the total lipid content, or the lipid and fatty acid class compositions were not influenced (Supplementary fig.6a), stearic acid (FA 18:0) was significantly reduced in pmembG6PD cells, accompanied by a 10% increase in the downstream ether lipid PE P-36:1 (Supplementary fig.6b).”